# Multicore fiber optic imaging reveals that astrocyte calcium activity in the mouse cerebral cortex is modulated by internal motivational state

Yung-Tian A. Gau[1] ✉, Eric T. Hsu[1,6], Richard J. Cha[2,6], Rebecca W. Pak[3,6], Loren L. Looger[4], Jin U. Kang[2,5] ✉ & Dwight E. Bergles [1,5] ✉

Astrocytes are a direct target of neuromodulators and can influence neuronal activity on broad spatial and temporal scales in response to a rise in cytosolic calcium. However, our knowledge about how astrocytes are recruited during different animal behaviors remains limited. To measure astrocyte activity calcium in vivo during normative behaviors, we utilize a high-resolution, long working distance multicore fiber optic imaging system that allows visualization of individual astrocyte calcium transients in the cerebral cortex of freely moving mice. We define the spatiotemporal dynamics of astrocyte calcium changes during diverse behaviors, ranging from sleep-wake cycles to the exploration of novel objects, showing that their activity is more variable and less synchronous than apparent in head-immobilized imaging conditions. In accordance with their molecular diversity, individual astrocytes often exhibit distinct thresholds and activity patterns during explorative behaviors, allowing temporal encoding across the astrocyte network. Astrocyte calcium events were induced by noradrenergic and cholinergic systems and modulated by internal state. The distinct activity patterns exhibited by astrocytes provides a means to vary their neuromodulatory influence in different behavioral contexts and internal states.

Astrocytes are ubiquitous and essential components of neural circuits in the mammalian CNS that are organized in a grid-like manner in the parenchyma, with each cell extending highly ramified processes to allow extensive interactions with neurons, glial cells, and the vasculature[1]. The fine lamellar protrusions that project from their processes separate and often encircle synapses, creating a "tripartite" structure with pre- and postsynaptic elements[2]. It has been estimated that each astrocyte contacts an enormous number of synapses (about 90,000 in rodents and 2,000,000 in humans)[1,3], allowing each cell to serve as a hub to integrate input from diverse sources to modulate the physiological output of neurons on broad spatial and temporal scales[4–9]. This functional integration of astrocytes is enabled in part by their expression of a diverse array of metabotropic receptors that can increase intracellular levels of calcium and other second messengers, triggering pleotropic effects ranging from regulation of glycolysis[10] to release of "gliotransmitters" that can influence neuronal activity

[1]The Solomon H. Snyder Department of Neuroscience, Johns Hopkins University, Baltimore, MD, USA. [2]Department of Electrical and Computer Engineering, Johns Hopkins University, Baltimore, MD, USA. [3]Department of Biomedical Engineering, Johns Hopkins University, Baltimore, MD, USA. [4]Howard Hughes Medical Institute, Department of Neurosciences, University of California San Diego, La Jolla, CA, USA. [5]Kavli Neuroscience Discovery Institute, Johns Hopkins University, Baltimore, MD, USA. [6]These authors contributed equally: Eric T. Hsu, Richard J. Cha, Rebecca W. Pak. ✉e-mail: gauy@upstate.edu; jkang@jhu.edu; dbergles@jhmi.edu

patterns and synaptic strength[9,11–13]. Recent single-cell mRNA sequencing studies have revealed that individual astrocytes exhibit distinct molecular characteristics and that translation occurs in their highly ramified processes, creating a framework for regionally-specific responses to global neuromodulation[3,11,14–18]. Although functional validation of this regional specification has started to emerge, many questions remain to be answered regarding the spatial extent and timing of their activity during different behaviors[19], the mechanisms that control their responses[20], and the consequences of their distinct activity patterns on local circuitry[21]. Understanding the complex role of astrocytes during different behavioral states has been limited by a paucity of information about the spatial and temporal dynamics of astrocyte activity during distinct voluntary behaviors.

Neuromodulatory systems are vital for our capacity to adapt and adjust behaviors based on internal states and changes in environment. In particular, these systems provide the means to register internal states, such as starvation versus satiety, or somnolence versus wakefulness. They can also convey information about external cues, such as the emergence of nutritional resources or unexpected rewards and novelties to enable adaptive behaviors[22–25]. As astrocytes are a direct target of neurotransmitters, including norepinephrine (NE), dopamine (DA) and acetylcholine (ACh)[13,26–30], and can be coupled into functional units[31], with access to many thousands of synapses, they are well-positioned to act both as an amplifier through which sparse neuromodulatory inputs convey contextual changes, and as a gain modulator to fine-tune local circuit output[6,7,13,32].

Understanding the relationship between the engagement of astrocytes and animal behavior requires the ability to monitor astrocyte activity patterns while animals are engaged in physiologically relevant behaviors. This has now been realized through the development of genetically encoded calcium sensors, cell-specific molecular targeting, and in vivo brain imaging. Nonetheless, in vivo imaging in the mouse brain commonly requires head immobilization to limit motion artifacts and enable use of heavy and complex imaging objectives. This approach is suitable for delivery of controlled stimuli and stereotyped training paradigms; however, restraining precludes full recapitulation of the complex behaviors that an animal performs in adapting to different contexts[33,34] and may be confounded by heightened arousal and stress. While head-mounted mini-microscopes have been widely used to visualize neuronal activity in freely moving animals, application of this approach has been more difficult for astrocytes[34], with most information about astrocyte calcium dynamics in freely behaving mice obtained using photometry rather than cellular imaging[26,30]. Astrocytes present unique challenges for in vivo imaging studies, due to their unique morphological features and physiological properties. In particular, they exhibit dramatic structural and physiological changes to brain injury, a transformation termed reactive astrocytosis[5,35], which is induced when micro-lenses used in mini-microscopes are implanted into the brain. This issue is further complicated by the scattering nature of brain tissue (and particularly at scar boundaries), making it difficult to image beyond the layer of reactive astrocytosis[36]. Moreover, the dense packing and highly ramified, thin processes of astrocytes place acute demands on the resolving power of imaging systems[1,37]. Resolution of fluorescence changes induced by genetically encoded calcium sensors is also more difficult in astrocytes, as their calcium transients arise predominantly through internal store mobilization, which are generally smaller events than those in neurons arising though action potential-dependent gating of calcium channels[38,39]. Although astrocyte metabotropic receptor-induced calcium transients in astrocytes are more prolonged, their small amplitude necessitates efficient photon capture, which is constrained by the small lenses employed in these devices.

To enable visualization of astrocyte calcium changes within the cerebral cortex of freely moving animals, we optimized a widefield, long working distance fiber optic imaging system capable of resolving cellular changes in fluorescence emitted by genetically encoded calcium indicators in vivo. Taking advantage of spatial-division multiplexing optics, this approach uses a multicore optical fiber coupled to miniature lenses that provide sufficient working distance to be mounted outside of a cranial window, minimizing brain injury. This platform enables repetitive imaging over long periods to delineate the activity patterns of both individual astrocytes and multicellular astrocyte networks in the upper layers of the cerebral cortex, which receives extensive neuromodulatory input. Using this approach, we defined the spatiotemporal dynamics of astrocytes during diverse behaviors, demonstrated that noradrenergic and cholinergic efferents coordinate to recruit astrocytes during various levels of arousal and attention, and showed that the detection threshold of astrocytes to a stimulus can be altered by internal state. This imaging platform provides a means to study how regulation of astrocyte physiology by neuromodulators influences activity in discrete circuits to modify information processing and behavior.

## Results

### Multicore fiber optic imaging allows visualization of astrocyte activity in vivo

Astrocytes in the upper layers of the cerebral cortex exhibit robust increases in calcium in response to behaviorally induced changes in neural activity[40,41], providing an opportunity to assess astrocyte network activity in freely moving animals using minimally invasive imaging through implanted optical windows. To define the behavioral contexts during which astrocytes are activated, we optimized a flexible fiber optic imaging system[40,42] to provide both the cellular resolution necessary to monitor fluorescence changes from individual astrocytes, and a field of view wide enough to visualize networks of astrocytes in these circuits. Imaging was achieved using a spatial-division multicore optical fiber in which 30,000 individual quasi-single mode glass fiber cores are assembled into a circular array <1 mm in total diameter with an inter-fiber spacing of 2.1 μm. Flexible multicore fibers have been previously exploited for photometric analysis and low-resolution, low signal-to-noise ratio detection of fluorescence changes in the brain[40]. To perform cellular resolution imaging at a distance, so as to avoid penetrating injury and tissue compression that occurs when imaging devices with a short focal distance, such as gradient-index lenses[14,35,43], are inserted directly into the brain, we coupled the end of the multicore fiber to two miniature aspheric lens doublets, together serving as an objective lens, achieving a numerical aperture of 0.58, yet retaining a working distance sufficiently long (2.5 mm) to allow imaging through a cranial window (Fig. 1a, b). Fluorescence imaging was performed using a 473 nm diode laser as an excitation source and a charge-coupled device (CCD) camera as a detector (see "Methods"), a configuration that achieved a lateral resolution of 2.8 μm and an axial resolution of 24 μm (Fig. 1c). Although used in an epifluorescence imaging mode, the small 2.1-μm aperture of each fiber enhances optical sectioning by acting as a pinhole to minimize detection of light from outside the focal plane. An additional advantage of this system is that most optical components (e.g., laser, camera, mirrors) can be placed remotely on a vibration isolation table using commodity components. Therefore, concessions in performance do not need to be made to minimize weight, as only the fiber lens array is attached to the head of the animal, and optics can be readily re-configured for alternative imaging methods, such as dual-spectral reflectance imaging (see below).

The imaging performance of this fiberscope was benchmarked against a commercial confocal microscope. Images of cortical astrocytes expressing the genetically encoded calcium indicator GCaMP3 (*GLAST-CreER; Rosa26-lsl-GCaMP3* mice; astro-GC3 mice)[40] in fixed brain slices were collected using a conventional confocal microscope and the fiberscope. This comparison revealed that the multicore system can resolve individual astrocytes and their processes (white

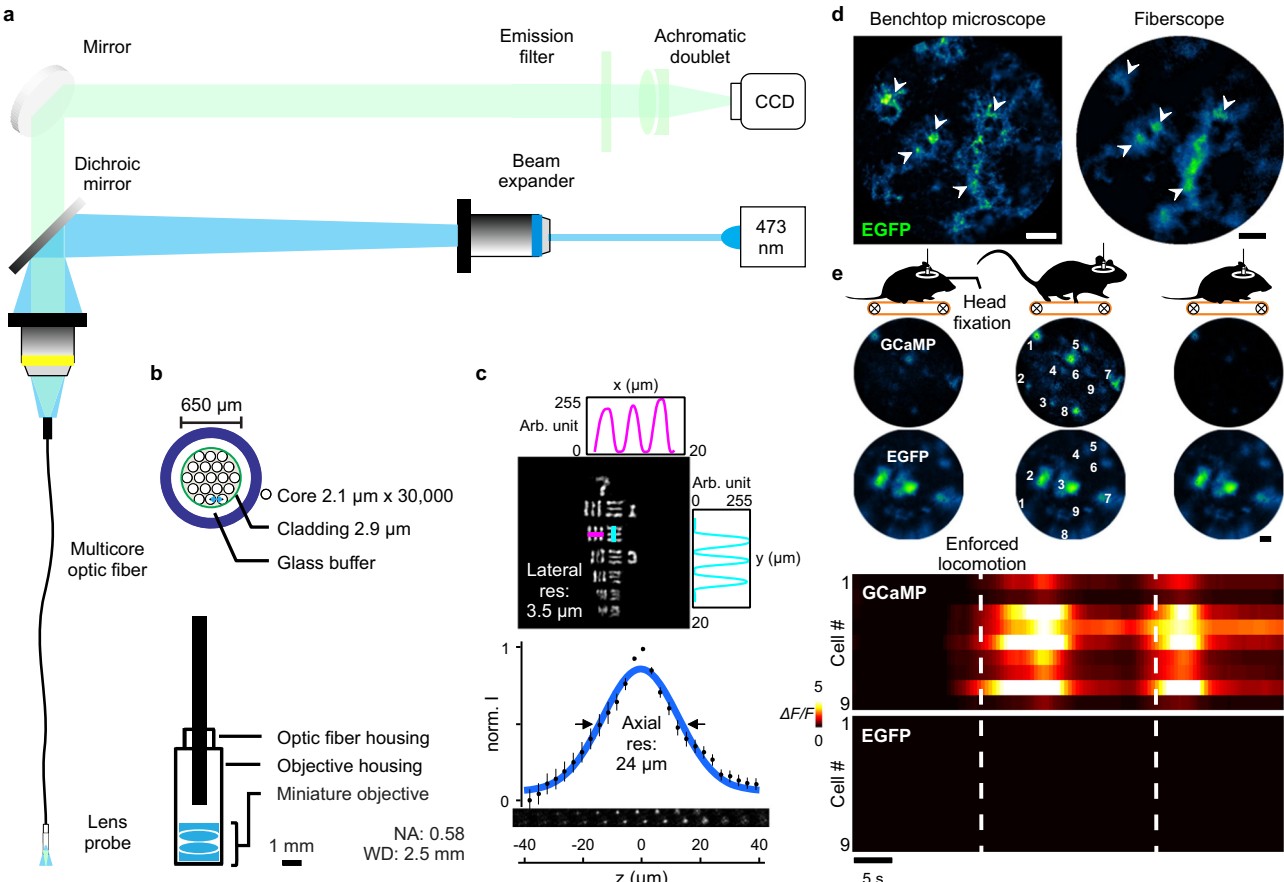

**Fig. 1 | An optimized fiberscope for imaging astrocyte calcium dynamics in vivo. a** Optical layout of the fiberscope imaging system. **b** Cross-section of multicore optical fiber bundle (top) and miniature objective configuration (bottom). **c** Resolution of the multicore fiber-miniature objective: 3.5 μm lateral resolution as in group 7 element 2 of the USAF target (top) and 24 μm axial resolution determined by the point-spread function of imaged microbead (bottom). **d** Comparison between a standard confocal microscope and the fiberscope imaging system using fixed tissue immuno-stained for EGFP. The white arrowheads denote individual astrocytes. Scale bar is 50 μm. **e** In vivo imaging using the fiberscope in head-fixed mice expressing either GCaMP or EGFP, and undergoing forced locomotion. Note that locomotion-induced increases in fluorescence were observed only in mice expressing GCaMP, indicating that these changes were not due to brain motion or other artifacts. Scale bar is 50 μm.

arrowheads, Fig. 1d), providing resolution similar to the benchtop system. To assess the performance of this system when imaging astrocytes in vivo, a micromanipulator was used to position the tip of the fiberscope over a cranial window implanted above the primary visual cortex (V1) of head-immobilized mice that expressed either GCaMP3 (astro-GC3) or EGFP (*GLAST-CreER; RCE:loxP* mice; astro-EGFP mice) selectively in astrocytes. We focused on V1 astrocytes to reduce the complexity of sensory-evoked inputs while studying neuromodulatory influences on astrocytes during complex behaviors. Astro-EGFP mice served as a control for artifacts arising from movement or microenvironmental fluctuations, such as pH. In astro-EGFP mice the fiberscope was able to resolve single EGFP-expressing cells in V1 ~75 μm from the surface, and in astro-GC3 mice it was able to detect GCaMP3 fluorescence within astrocytes in the imaging field (Fig. 1e). Mice were induced to walk on a motorized treadmill (scheme, Fig. 1e), a condition known to elicit a global rise in calcium within cortical astrocytes due to the local release of norepinephrine (NE) that accompanies enhanced arousal[27,40,44]. Astrocytes throughout the imaging field in astro-GC3 mice exhibited transient increases in fluorescence (Fig. 1e) time-locked to the onset of ambulation, consistent with NE inducing global release of calcium from intracellular stores. Under comparable imaging conditions, no appreciable fluorescence changes were observed when astro-EGFP mice were induced to walk, indicating that the fluorescence changes in astro-GC3 mice arose though changes in intracellular calcium rather than motion-associated artifacts. Together, these results

show that this lensed multicore fiber imaging system has sufficient sensitivity and resolution to monitor calcium levels within individual astrocytes in vivo.

## Imaging individual astrocyte activity in freely moving mice

To allow imaging of cortical astrocyte activity in freely moving mice, we used a rigid, adjustable metal collet to secure the lensed-fiberscope to the mouse skull (Fig. 2a)[45]. This mount allowed fine positioning of the fiber relative to the cranial window in both x–y and z planes, and facilitated longitudinal analysis of astrocyte activity over multiple days, as the fiberscope could be removed and readily repositioned to image the same focal plane, using the surface blood vessels as fiducial landmarks (Supplementary Fig. 1). Astrocyte activity was monitored simultaneously with animal behavior by synchronizing the output of the fiberscope CCD with the output of two orthogonally-oriented near infrared (NIR) cameras placed outside the arena (Fig. 2b). Animals that were allowed to roam freely in the arena with the fiber tethered exhibited similar behaviors as those that were untethered, in terms of their preferred location within the cage, the total distance moved (untethered: 0.068 ± 0.009 km, *n* = 7; tethered: 0.079 ± 0.008 km, *n* = 7; two-sample *t* test, *t* (12) = 0.88, *p* = 0.40), and the speed of their movement (untethered: 5.7 ± 0.27 m/s, *n* = 7; tethered: 6.7 ± 0.37 m/s, *n* = 7; two-sample *t* test, *t* (12) = 2.1, *p* = 0.06) (Fig. 2c). Tethered mice also demonstrated diverse spontaneous behaviors comparable to those of the untethered mice (difference between the two groups:

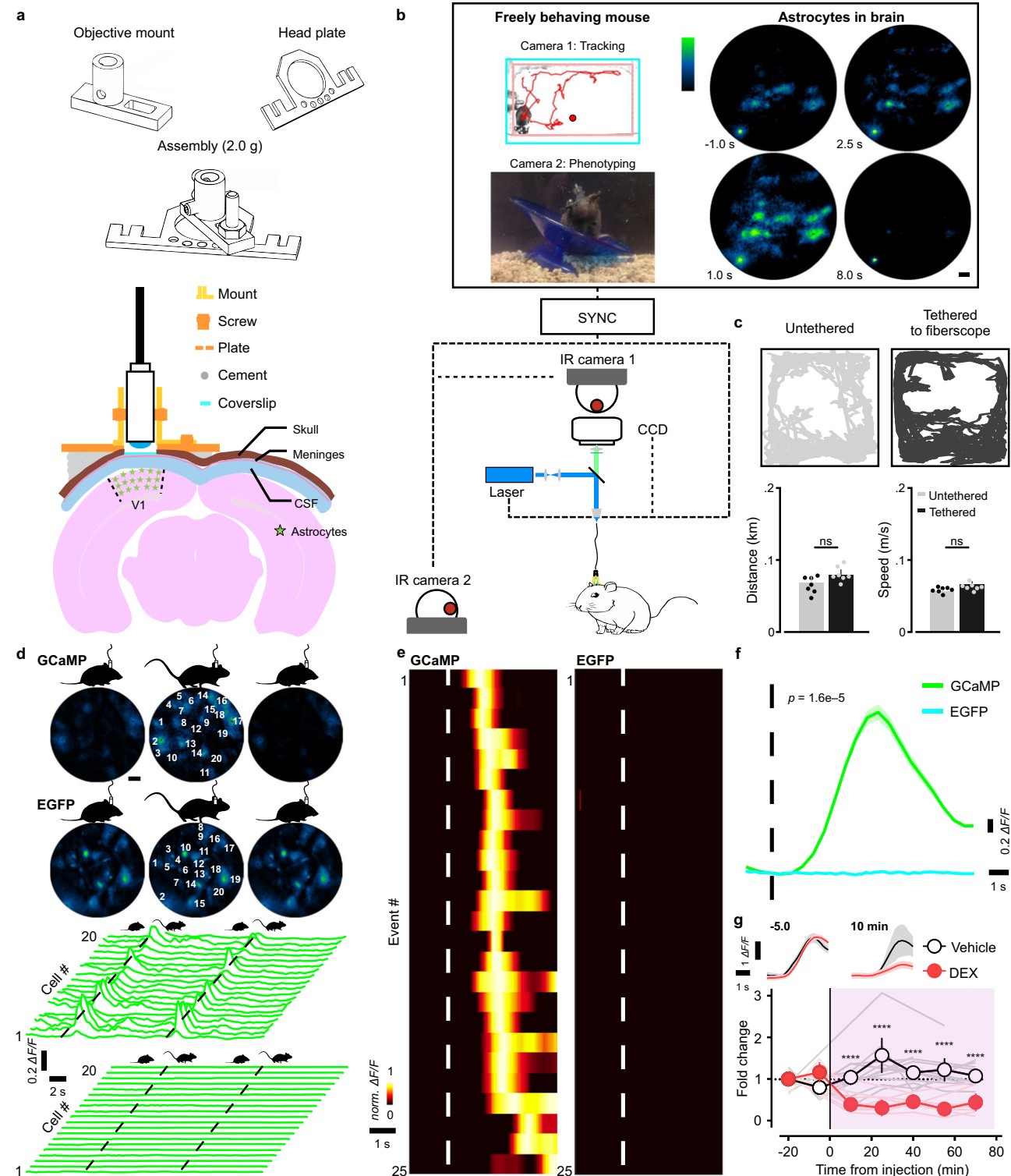

**Nature Communications**| (2024)15:3039

$-1.67 \pm 1.84$, 95% CI, $[-5.34, 2.01]$, $n = 7$ for each group; two-way ANOVA, $F(5, 72) = 1.51$, $p = 0.20$) (Supplementary Fig. 2).

To determine if this configuration was stable enough to allow astrocyte imaging in freely moving mice, we monitored fluorescence changes in both astro-EGFP and astro-GC3 mice. When animals voluntarily transitioned from a quiescent to active state (Q-A transition), astrocytes in astro-GC3 mice exhibited consistent increases in fluorescence (Fig. 2d), which occurred within ~1 s after the onset of movement (dashed black line, Fig. 2d), as shown by raster plots of across-cell fluorescence changes aligned to movement onset (dashed white line, Fig. 2e), both for individual trials and when averaged across trials and animals ($2.59 \pm 0.59$ $\Delta F/F$ at 90% rise; $n = 6$) (Fig. 2f). In contrast, when similar experiments were performed in astro-EGFP mice, Q-A transitions did not elicit changes in astrocyte fluorescence ($0.03 \pm 0.01$ $\Delta F/F$ at the time of 90% rise in astro-GC3 mice; $n = 5$) (Fig. 2e, f). Astrocyte GCaMP activity associated with Q-A transitions was strongly suppressed when animals were administered the α2-adrenoceptor agonist dexmedetomidine (DEX, 3 μg/kg intraperitoneal), which binds to presynaptic receptors to inhibit release of NE from noradrenergic nerve terminals (difference in fold change at

**Fig. 2 | Imaging of astrocyte calcium in freely behaving mice. a** Diagram of fiberscope-mouse brain coupling. Top: three-dimensional render of the mount for securing the fiberscope to the mouse skull. Bottom: minimally invasive optics-animal coupling for imaging of cortical astrocytes through cranial windows. **b** Outline of the configuration used to simultaneously monitor astrocyte calcium signals and mouse behavior. Output of the brain imaging CCD camera were synched with the two behavior-recording IR cameras. Scale bar is 50 μm. **c** Plots of mouse behavior over one hour showing that tethering to the optical fiber does not substantially change their behavior or the total distance they travel (untethered: 7 mice vs. tethered: 7 mice) (two-sample *t* test, ns: not significant). **d** Imaging fluorescence transients of 20 cells in a freely moving GCaMP and an EGFP animal. Corresponding traces quantified increases in GCaMP fluorescence (calcium increases) in astrocytes within the visual cortex during voluntary

locomotion, in contrast to those of the EGFP animals where no significant changes were found. Scale bar is 50 μm. **e** Rank-ordered population (average of 20 cells) fluorescence intensity plot for 25 voluntary locomotion events for mice expressing GCaMP or EGFP. **f** Summary data for locomotion-induced fluorescence changes in astrocytes in mice expressing GCaMP (5 animals) or EGFP (6) (two-way mixed rank test, ANOVA-type). **g** Plot of fold-increase in astrocyte fluorescence (calcium) during voluntary locomotion in mice injected with vehicle (1.14 ± 0.19, 6 animals) vs. dexmedetomidine (0.33 ± 0.06, 9 animals) (two-way repeated measurement ANOVA with post hoc Sidak's multiple comparison, ****$p < 0.0001$). Global locomotion-induced calcium transients are triggered by norepinephrine release, as they are strongly attenuated by dexmedetomidine, which inhibits the release of NE. Data are presented as mean ± s.e.m. Source data are provided as a Source Data file.

10 min: 0.81 ± 0.17; 95% CI, [0.3, 1.3]; Vehicle: $n = 6$, DEX: $n = 9$; two-way repeated measurement ANOVA, $F_{(1, 90)} = 111$, $p < 0.0001$) (Fig. 2g), indicating that astrocyte calcium levels are consistently elevated by NE during spontaneous arousal.

Neuronal and glial cell activity can cause local changes in blood volume, blood oxygenation, and changes in the optical properties of the brain[46,47], which alter the transmission of light and influence the detection of fluorescence signals[48,49]. To assess the degree to which responses recorded from astro-GC3 mice are influenced by changes in neurovascular-associated absorbance, during in vivo imaging we measured the reflectance signal at 473 nm and 523 nm, corresponding to the peak excitation/emission wavelengths used for GCaMP imaging (Supplementary Fig. 3a). Measured GCaMP fluorescence signals were then corrected to compensate for these hemodynamic changes (Supplementary Fig. 3b)[48,49]. Although Q-A transitions were associated with small reflectance changes at both wavelengths (Ex 473 nm, Em 523 nm) (Supplementary Fig. 3c), consistent with findings in head-restrained mice[50], these changes in light absorbance had a minimal effect on the amplitude and time course of GCaMP fluorescence signals (Supplementary Fig. 3c). Hemodynamic-dependent crosstalk in GCaMP fluorescence signals depends on the path length travelled by photons. As most GCaMP fluorescence in this preparation arises from astrocytes in the upper cortical layers, photons travel a shorter path to the fiber end, and are therefore contaminated exponentially less by hemodynamics than those from deeper layers[48,49]. Thus, GCaMP fluorescence changes in subsequent experiments were not adjusted to account for these hemodynamic reflectance changes, unless otherwise noted. Together, these studies indicate that the lensed multicore fiberscope can resolve fluorescence signals from individual astrocytes during animal movement and is sufficiently sensitive to detect increases in GCaMP3 fluorescence initiated by physiological changes in cytosolic calcium.

### Longitudinal imaging reveals changes in astrocyte activation during active and inactive phases

Astrocytes have been shown to influence sleep homeostasis[34,37]. In particular, sleep pressure, a phenomenon in which the motivation for sleep increases progressively during wakeful periods, is thought to be encoded by astrocyte calcium activity[29,51]. This accumulation of sleep need leads to release of astrocytic sleep-promoting substances such as adenosine[52]. To understand how astrocytes are engaged to control this behavior, there is a pressing need for longitudinal, in vivo monitoring of astrocyte activity with cellular resolution during endogenous sleep-wake transitions. Therefore, we imaged freely moving mice during different phases of the circadian cycle (Fig. 3a). To minimize photobleaching, the field-of-view was illuminated for 1 s every 60 s (Fig. 3a). Sleep was inferred when the animal was immobile for >40 s, which has previously determined to have 95–99% correlation with simultaneous electroencephalography- and electromyography-defined sleep[53,54]. Mice carrying the fiberscope exhibited gradual changes in movement over 24 h, evident when

individual activity profiles were aligned to the most active period (Fig. 3b), suggesting that this imaging approach allowed preservation of intrinsic circadian behavior. We then monitored individual astrocyte calcium levels over 24 h to define their spatiotemporal activity across the intrinsic sleep/wake cycle (Fig. 3c). Within individual animals, the frequency of global astrocyte events (events that occurred in astrocytes throughout the imaging field) tracked with the phase of animal activity (Fig. 3d), perhaps reflecting overall levels of neuromodulatory output. As expected, astrocyte calcium transients occurred more frequently during the active phase (0.21 ± 0.03, 95% CI, [13.7, 28.1], $n = 7$), than the inactive phase (0.07 ± 0.02, 95% CI, [2.89, 11.0], $n = 7$) (Wilcoxon rank-sum test, $U = 47.0$, $p < 0.001$) (Fig. 3e). Similarly, when astrocytes were active, it was more likely that the animals were also active (Fig. 3f) (actual: 31.7 ± 0.1, 95% CI, [25.6, 38.7], $n = 1000$ vs. random: 24.0 ± 0.1, 95% CI, [19.9, 28.7], $n = 1000$) (two-sample K–S test = 0.8480, $p < 0.001$). These findings show that astrocyte calcium event frequency varies predictably with animal activity levels according to the endogenous rhythm, providing a means for astrocytes to integrate information about cumulative sleep and wake periods.

### Astrocytes are active during exploration rather than maintenance behaviors

A major advantage of imaging in freely moving animals is the ability to define which of a broad spectrum of behaviors are associated with astrocyte activity. To assess which behaviors were temporally correlated with changes in astrocyte calcium signaling, we monitored astrocyte fluorescence in the visual cortex of astro-GC3 mice as animals engaged in a variety of normal behaviors in addition to the Q-A transitions, such as grooming, nesting, exploring, drinking, and feeding (Fig. 4a, b). To detect and annotate these behaviors in NIR video sequences, we used a machine learning-based detection program[55] that was 94 ± 3.0% accurate in categorizing these behaviors, as independently assessed when blinded to the computer-aided categorization (see "Methods") (Fig. 4a). Although transient increases in astrocyte calcium levels were occasionally observed during diverse behaviors, Q-A transitions and exploratory behaviors such as rearing were most prominently correlated with enhanced astrocyte activity (Fig. 4b, c). These studies are consistent with the conclusion that astrocyte calcium transients occur most prominently during periods of enhanced arousal, though not exclusively during locomotion.

### Diverse exploration in novel environments activates astrocyte networks

Animals often exhibit characteristic exploratory behavior when encountering a new environment[56,57]. For example, mice respond to new environments by rearing on their hind legs to gather information about their surroundings[58], a diverse explorative behavior associated with spatial/contextual learning[59]. We therefore investigated whether calcium signaling was induced in astrocytes when mice explored novel

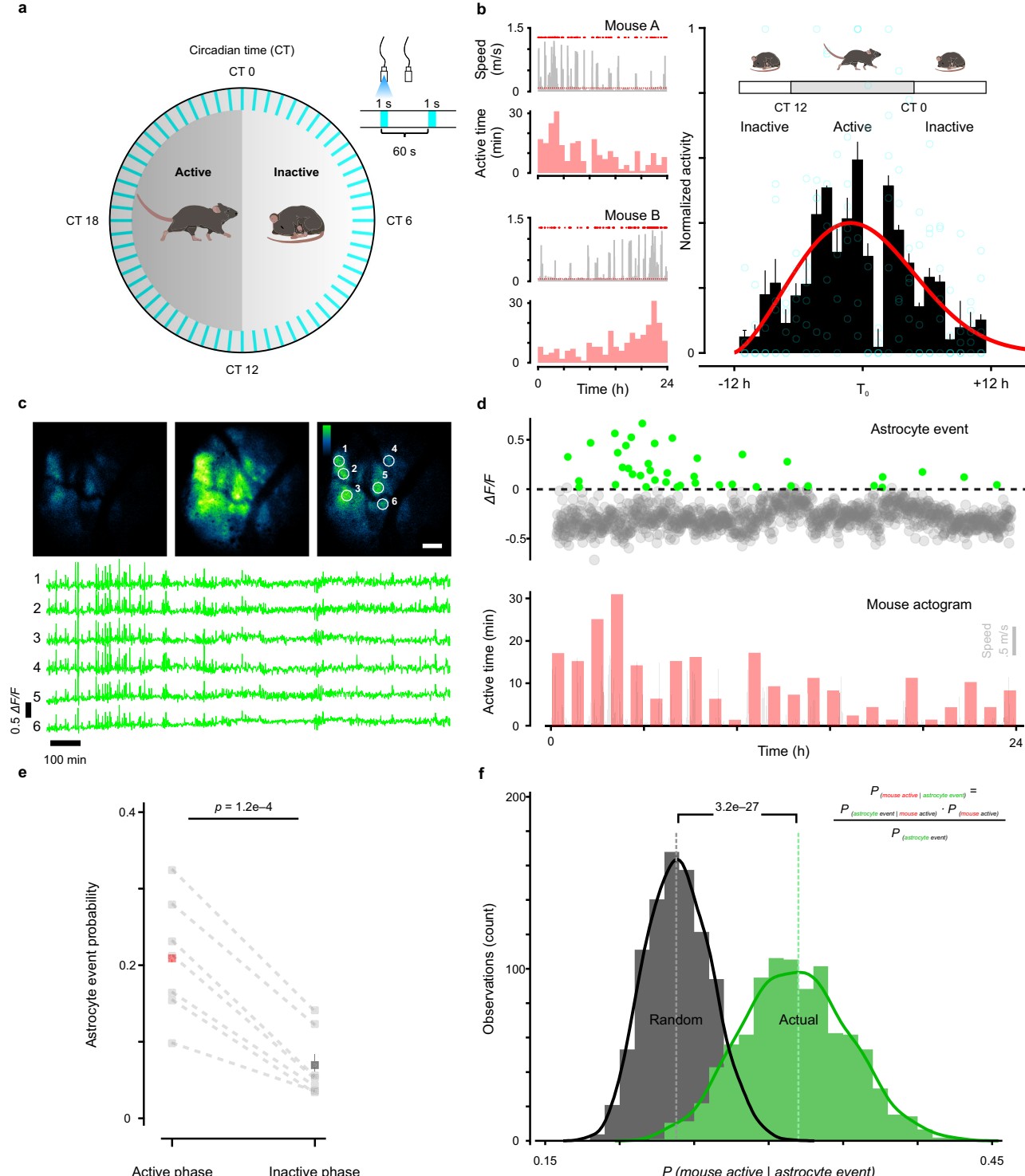

**Fig. 3 | Freely moving imaging across 24 h reveals diurnal variation in astrocyte activity. a** Scheme of astrocyte monitoring in a behaving animal across 24 h. **b** Left: two examples of activity level variations following the intrinsic rhythms of individual mice over 24 h. Right: phase-aligned hourly activity across six animals showed robust preservation of the endogenous cycle (red: probability distribution with cubic spline fit). **c** Top: representative images from a 24 h session showing astrocytes with baseline, maximal, and intermediate activity, respectively (scale bar is 50 μm). Bottom: corresponding $\Delta F/F$ traces of the six cells highlighted in the image above. **d** Astrocyte calcium transients (green: astrocyte activation) appeared to follow the diurnal variation of mouse activity (bottom). **e** Astrocytes of seven animals show more transients during the active phase of the diurnal cycle (red: 0.21 ± 0.03) than during the inactive phase (black: 0.07 ± 0.02) (Wilcoxon rank-sum test). **f** Astrocyte transients are indicative of animal activeness. Black denotes the Bayesian estimated probability if activity of the mouse were a random process; green represents the calculated probability from actual data (bootstrapped 1000 times) (two-sample K–S test). Data are presented as mean ± s.e.m. Source data are provided as a Source Data file.

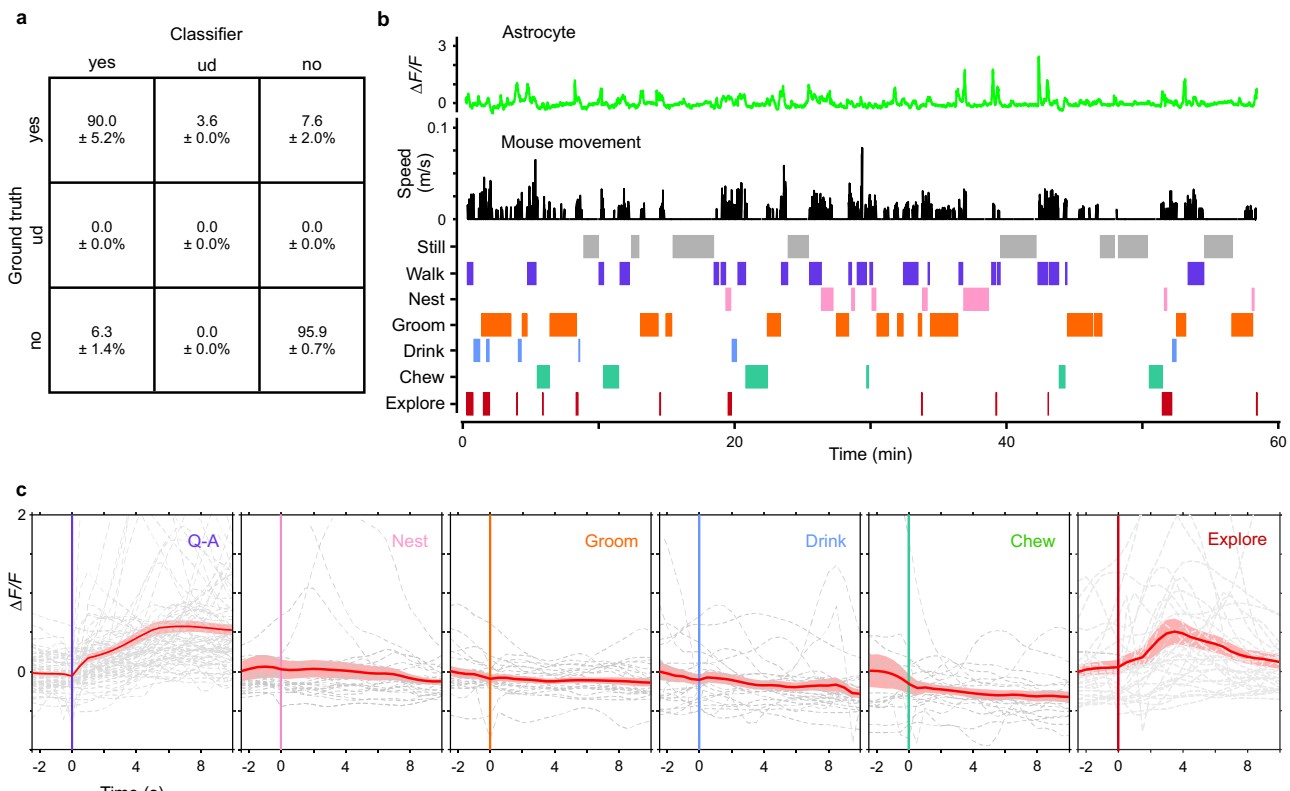

**Fig. 4 | Astrocytes exhibit distinct calcium responses during different behaviors. a** Human-provided ground-truth labels over 50,000 frames yielded 93.9 ± 3.0% accuracy for the machine learning classifier in categorizing behaviors (ud: undetermined). **b** Longitudinal monitoring of astrocyte calcium changes (top) and mouse movements (middle) during spontaneous behaviors (bottom). **c** Plots of astrocyte calcium transients aligned to the onset of behavioral events. Gray lines are individual events and red lines are the average responses across six animals. Note that the most consistent activation occurred at Q-A transitions and during exploration, such as rearing. Data are presented as mean ± s.e.m. Source data are provided as a Source Data file.

areas[60]. To quantify astrocyte activity explicitly during this behavioral context, we identified rearing events from movies as mice explored the arena by continuously computing body length (see "Methods"), which becomes highly extended during rearing. This algorithm reliably distinguished rearing from other movements in the arena (Fig. 5a, c), allowing GCaMP fluorescence signals in astro-GC3 mice to be aligned to each rearing event (Fig. 5b, c). Analysis of these events revealed that rearing was consistently associated with increases in astrocyte calcium levels, visible as transient increases both in the GCaMP fluorescence of individual astrocytes in an imaging field (Fig. 5b), and in the average cell fluorescence change across animals (Fig. 5d); in contrast, animals that were continuously active, but not rearing, did not exhibit consistent changes in astrocyte calcium (Fig. 5c, d). Correction of these fluorescence measurements for hemodynamic changes confirmed that rearing events were consistently associated with increases in astrocyte calcium (Supplementary Fig. 3c); however, the amplitude of the GCaMP fluorescence rise was not obviously correlated with either the duration of the rearing event ($\Delta t_{rear}$) or the magnitude of body extension ($\Delta h_{rear}$) (Fig. 5e).

To understand which aspects of this complex behavior were most closely linked to astrocyte calcium changes, the peak increase in GCaMP fluorescence was aligned to the onset and offset of the behavior and rank-ordered relative to delay time (Fig. 5f). Because the peak fluorescence change occurred with a latency of 2.45 ± 0.12 s after onset ($n = 120$), and rearing events followed a relatively consistent time course lasting 1.53 ± 0.07 s ($n = 120$), the maximal astrocyte calcium response occurred well after onset of the behavior (Fig. 5f, g). When rearing events lasted longer, there was a concomitant increase in the delay to the peak in astrocyte calcium, visible when onset and offset time were included on the same rank-

order raster plot (Fig. 5g), suggesting that astrocyte activity is more closely linked to the cessation rather than the onset of this behavioral sequence. The consistency in the timing between this behavioral sequence and peak astrocyte calcium was assessed further by calculating the ratio of Fano factor (rFF, the ratio of the variance to the mean of the calcium transients among defined intervals), which is often used to describe the variance of action potentials in spike trains during in vivo recordings[61], in which lower rFF values indicate higher deviation from a random process[62,63]. During rearing events, the rFF deviated 5.7% from baseline upon rearing onset (Fig. 5h, $t_1$, from 100 ms to 900 ms after baseline), but declined as much as 21.6% at the offset (Fig. 5h, $t_1'$, from 100 ms to 900 ms after), indicating a closer association between astrocyte calcium and rearing offset. To better quantify this relationship and address the possibility of random association, we analyzed the distribution of latency from the peak astrocyte transient to the time of onset and offset using Lorenz plots with Gini coefficient[64], providing a measure of how these intervals vary from an equal distribution. The Gini coefficient was 0.24 for the onset latency distribution and 0.21 for the offset distribution (Fig. 5i). The difference in Gini coefficients indicates that the associations are not random and that there is a closer relationship between the peak and event offset than event onset. Moreover, the coefficient of variation (CV) of latencies to the peak astrocyte response was slightly higher for the onset (0.32 ± 0.05, 95% CI, [0.23, 0.42], $n = 25$) than for the offset (0.30 ± 0.05, 95% CI, [0.20, 0.39], $n = 25$) (Wilcoxon signed-rank test, $p = 0.02$) (Fig. 5i, inset). Given the slow response dynamics and variable time to peak of the astrocyte calcium responses, we repeated the same latency analysis referenced to the onset (10% of peak calcium rise) (Supplementary Fig. 4a). Consistent with the peak analysis, the latency to rearing offset was

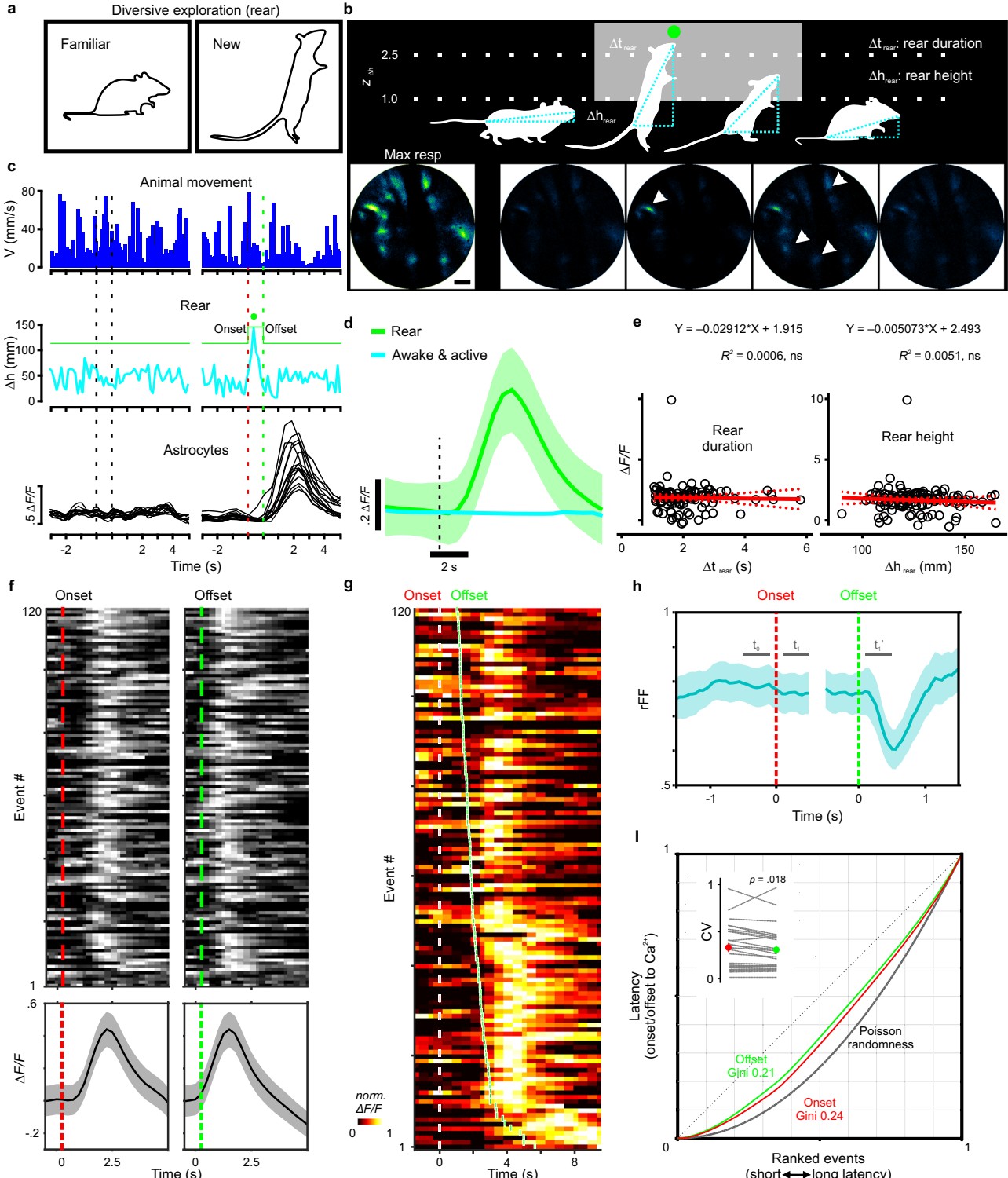

**Fig. 5 | Diversive exploration (rearing) in environmental novelty engages astrocyte networks. a** Diagram of encouraged rearing, a selective form of diversive exploration, in novel spatial environments. **b** Top: quantitative analysis for capturing and parameterizing each rearing event. Bottom: representative GCaMP fluorescence images in astro-GC3 mice during rearing. Scale bar is 50 μm. **c** Traces of cortical astrocyte activities (black) when mice moved continuously (blue) and reared (cyan). **d** Calcium transients plotted across animals, comparing rearing to moving behaviors. **e** Transient magnitude was not correlated with duration of the rearing event ($\Delta t_{rear}$) or the extent of body extension ($\Delta h_{rear}$). **f** GCaMP fluorescence was aligned (heat map) and averaged (bottom trace) across events to the onset and offset of the behavior. **g** Onset-aligned and offset-ranked raster plot demonstrated a concomitant increase in the delay of the peak in astrocyte calcium when rearing

events lasted longer. **h** During rearing events, in comparison with baseline ($t_0$, denoting the final 900 to 100 ms before the event), the ratio of Fano factor (rFF) reduced as little as 5.7% after rearing onset ($t_1$, from 100 ms to 900 ms after), but declined by 21.6% after the end of offset ($t_1'$, from 100 ms to 900 ms after), indicating a closer association between astrocyte calcium and rearing offset. **I** Lorenz plots for the latency of the peak astrocyte calcium transients and the onset and offset times: Gini was 0.24 for onset latency distribution and 0.21 for offset. Inset: coefficient of variation (CV) of latencies to the astrocyte response was higher for the onset (red: $0.32 \pm 0.05$) than for the offset (green: $0.30 \pm 0.05$) in 25 mice (Wilcoxon signed-rank test). Data are presented as mean $\pm$ s.e.m. Source data are provided as a Source Data file.

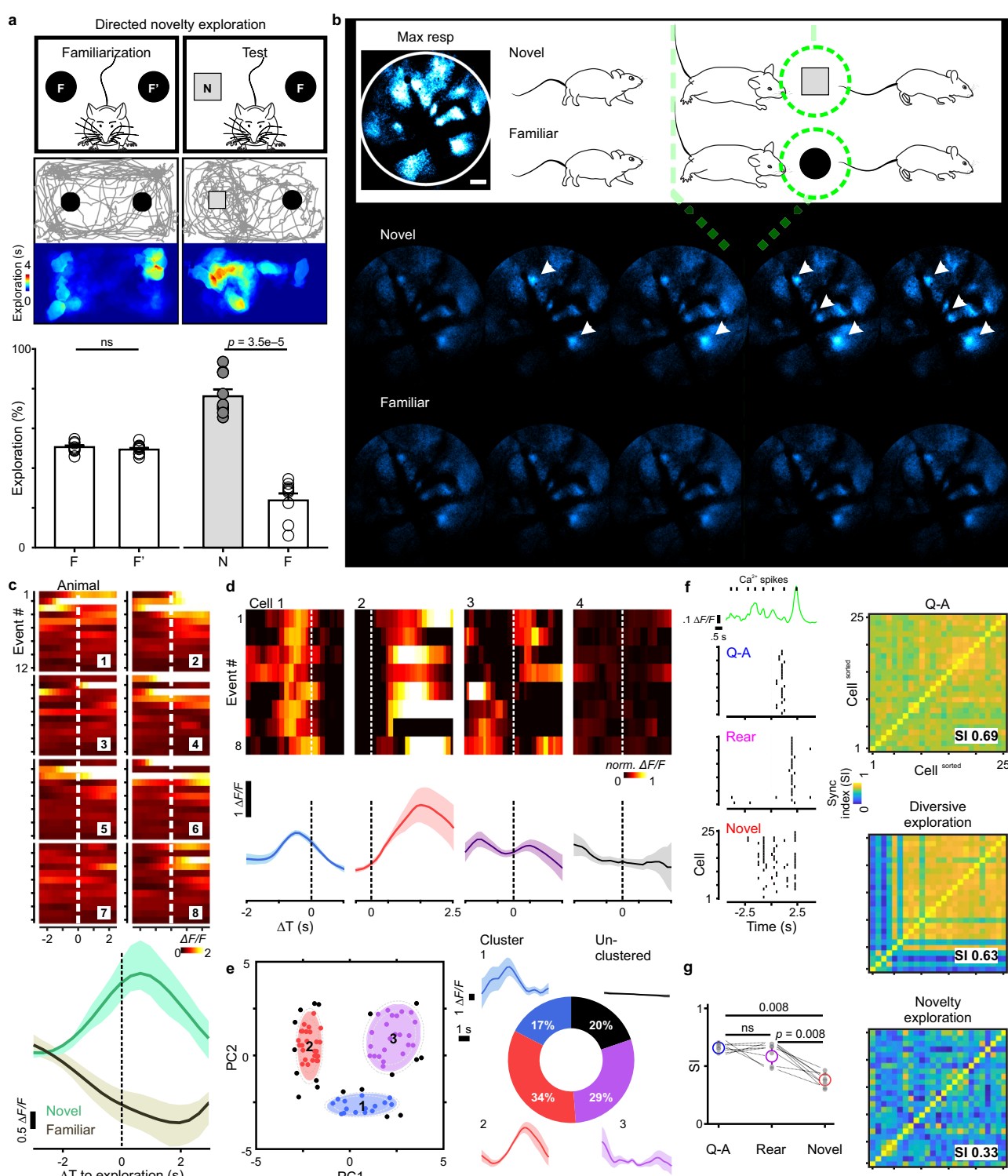

less variable in relation to the beginning of the calcium rise than to rearing onset, as shown in the Lorenz plots (Supplementary Fig. 4b, Gini: offset 0.12 vs. onset 0.13) and the CV of latencies (Supplementary Fig. 4c, offset: $0.16 \pm 0.03$, 95% CI, [0.10, 0.21] vs. onset: $0.18 \pm 0.03$, 95% CI, [0.12, 0.25], $n = 25$) (two-tail paired $t$ test, $p < 0.0001$). Thus, as compared to enforced locomotion and spontaneous Q-A transitions that elicited rapid elevation of calcium, in this exploratory behavior, astrocyte networks were consistently activated following the end of the behavioral sequence, indicating that astrocytes can be activated during discrete periods of complex behaviors.

## Astrocytes become desynchronized and adapt rapidly during inspective exploration

We further assessed astrocyte activity when animals inspected novel objects (Fig. 6a, scheme). Mice that demonstrated no more than 55% object bias in a familiarization session proceeded to the test session ($p = 0.48$ for percentage of time spent exploring two familiar objects, F or F') (Fig. 6a). During test sessions, mice were returned to the arena where one of the familiar objects had been replaced with a new object. All animals in test sessions spent more time exploring the new object (percentage of time spent exploring the familiar object, F: $24\% \pm 3.5\%$, 95% CI, [16%, 32%] or the new object, $N$: $76\% \pm 3.5\%$, 95% CI, [68%, 84%],

**Fig. 6 | Astrocyte networks desynchronized and adapted rapidly during inspective exploration. a** Scheme of novel object recognition paradigm (top) where a mouse preferentially traveled (trace) and spent time (color map) around the novel object. Bottom bar chart: establishment of paradigm across ten animals, shown by consistent exploration preference toward the novel object (paired *t* test, ns not significant). **b** Representative images (scale bar is 50 μm) of cortical astrocytes while animals explored the novel vs. the familiar object, with the onset of directed exploration indicated by green dashed lines. **c** In eight representative animals, population transients (across-cell average) were time-locked to the directed exploration toward the novel object, but not toward familiarity. Of note, levels of astrocyte engagement declined rapidly with repetitive interactions, indicating fast adaptation from event 1 to 12. **d** Top: heat map with Δ*F/F* normalized to the max (norm. Δ*F/F*, ranging from 0 to 1) during each event suggested that timing of calcium transients within individual cells was remarkably consistent in relation to

the behavioral sequence. Bottom: across-event calcium average of the individual cells. **e** Left: Gaussian mixture model with expectation maximization sorted principal component (PC1 and PC2) into three clusters (80.2% of 86 astrocytes) with the color edge at 95% likelihood (outer broken grey line: 99% likelihood). Right: within-cluster calcium averages suggested astrocytes of each cluster shared distinct temporal features. **f** Left: representative network of 25 cells during a Q-A transition, a diversive rearing exploration, and inspective novelty exploration (detected calcium transients plotted as black vertical raster). Right: synchronization indices to quantify the network difference visualized in the raster plots. **g** Across eight animals, inspective explorations consistently desynchronized cortical astrocytes, distinct from the homogeneity observed when mice entered a state of enhanced arousal during Q-A transitions and rearing (Friedman test with post hoc Dunn's multiple comparison, global *p* = 0.0011). Data are presented as mean ± s.e.m. Source data are provided as a Source Data file.

*n* = 10 animals) (paired *t* test, *t* (9) = 7.5, *p* < 0.0001) (Fig. 6a), indicating that they recognized novelty in the current setting (discriminability index d': 0.52 ± 0.07, 95% CI, [0.37, 0.68]) (one-sample *t* test, *t* (9) = 7.5, *p* < 0.0001). We then monitored calcium fluctuations in astrocytes within the visual cortex while animals explored either novel or familiar objects (Fig. 6b). The time of onset of directed exploration was defined as the moment the mouse came within 2.5 cm and oriented their head towards the object (Fig. 6b, green dashed lines). Inspective exploration of the novel object triggered small, rapidly rising and decaying calcium transients in subsets of astrocytes that often began before onset of interaction (Fig. 6b, Novel). In contrast, exploration directed toward a familiar object consistently failed to elicit detectable calcium increases in astrocytes (Fig. 6b, Familiar).

Quantifying full-field population dynamics across events for different animals (Fig. 6c, color maps) revealed that astrocyte calcium signaling increased consistently during initial interactions with a novel object (0.29 ± 0.08 Δ*F/F* at 90% rise; 95% CI, [0.04, 0.56], *n* = 8) (Fig. 6c), but not during interactions with a familiar object (−0.24 ± 0.06 Δ*F/F*; 95% CI, [−0.65, 0.14], *n* = 8) (Fig. 6c), with the two conditions being significantly different (two-way repeated measures ANOVA, *F* (1, 7) = 5.5, *p* = 0.03). The hemodynamic contamination was negligible (Supplementary Fig. 3c). In accordance with behavioral adaptation, astrocytes exhibited a consistent decrease in responsiveness with successive interactions with a novel object (compare responses during the initial events to the later in the color map of Fig. 6c); comparing the mean response (90% rise) of the last three events (−2.6 ± 0.7 Δ*F/F*, 95% CI, [−4.2, −1.0], *n* = 8) to that of the first three events (14.1 ± 5.9 Δ*F/F*, 95% CI, [0.2, 28.1], *n* = 8), revealed that the response of astrocytes decreased by 156 ± 20% (95% CI, [109%, 202%], *n* = 8). These findings indicate that cortical astrocyte activity is updated rapidly and adapts to changing behavioral contexts.

Unlike the coordinated astrocyte activity observed during Q-A transitions and rearing, astrocyte responses appeared much less correlated during directed novelty exploration (Fig. 6b). To determine if individual astrocytes exhibited consistent responses during this behavior sequence, we aligned the GCaMP traces from individual cells (Fig. 6d). This presentation revealed that despite different response profiles, within individual astrocytes the peak of calcium changes with respect to the time of object encounter seemed remarkably consistent across sessions. We attempted to cluster astrocytes according to the timing and magnitude of their peri-exploration calcium transients over eight consecutive trials. Applying principal component analysis to the data yielded a first component containing 76.6% of the variance, and a second containing 12.2% of the variance. A Gaussian mixture model was then fitted to the scores of these two components using an expectation-maximization algorithm (see "Methods"), which, by threshold of 95% likelihood, identified three main subpopulations of astrocytes (of 86 astrocytes analyzed) with shared response characteristics (Fig. 6e, Cluster 1–3). The average calcium transients for astrocytes were plotted for each cluster, revealing that 80.2% of

astrocytes maintained their distinct temporal features between trials (Fig. 6e, traces). The temporal differences in their responses enabled astrocytes to, in aggregate, spread their activity over the course of the exploration event, with different groups aligned to discrete behavioral contexts (Fig. 6b, e). Astrocytes that were not contained within the clusters (Fig. 6e, black) typically exhibited responses that were inconsistent from trial to trial. Since clustering was based on calcium changes associated with peri-inspective exploration, these unclustered cells may serve as reserve to encode alternative behavioral contexts, as proposed for neurons.

To further define individual cellular activities in diverse contexts, we plotted calcium events from a representative network of 25 astrocytes (Fig. 6f, detected calcium transients plotted as black vertical raster lines) during Q-A transitions, diversive rearing explorations, and directed novelty explorations (Fig. 6f). This analysis revealed that directed explorations were associated with less synchronized astrocytic activity (Fig. 6f, Novel), in contrast to the highly-synchronized calcium events observed when the mouse experienced a state of arousal (Fig. 6f, Q-A, synchronization index (SI) = 0.69), a phenomenon seen for all animals (Fig. 6g) (0.67 ± 0.02, 95% CI, [0.64, 0.71], *n* = 8). Desynchronization emerged when the mouse explored new surroundings (Fig. 6f), and became most prominent when its attention was directed toward a target (Fig. 6f) (Novel, SI = 0.33). This astrocyte desynchronization in directed/inspective exploration was significant across animals (Fig. 6g) (0.36 ± 0.03, 95% CI, [0.30, 0.43], *n* = 8) (Friedman test, *F* (9) = 12, *p* < 0.01). Although a form of exploration, rearing did not produce such marked desynchronization (Fig. 6f, g) (0.63 ± 0.04, 95% CI, [0.53, 0.73], *n* = 8), perhaps because rearing is often associated with increased vigilance/arousal[60]. For comparison, we also calculated the SI for behaviors not otherwise classified (0.17 ± 0.01, 95% CI, [0.15, 0.20], *n* = 8) and also for random time segments (0.20 ± 0.01, 95% CI, [0.17, 0.22], *n* = 8), which both exhibited minimal synchronization. Thus, astrocyte activity becomes increasingly correlated as the mice become more vigilant and engage with a task.

## Contextual encoding by cortical astrocytes is dependent on convergent neuromodulation

To explore the neural mechanisms responsible for the distinct response profiles of astrocytes in different behavioral contexts, we monitored changes in their responses after topical application of antagonists to the cortical surface (Fig. 7a, scheme). In accordance with previous studies (Fig. 2g, ref. 40), the α1 adrenoceptor antagonist prazosin reduced the amplitude of astrocyte calcium transients induced during Q-A transitions (Δ*F/F* after-before: aCSF 0.00 ± 0.04, *n* = 10 vs. prazosin 0.25 ± 0.09, *n* = 8) (Kruskal–Wallis, *H* = 18.4, *p* = 0.001) (Fig. 7a, Q-A); in some experiments, prazosin also slowed the rise of responses, but this did not reach significance (*p* = 0.06). Astrocyte responses during rearing events were similarly suppressed by prazosin (Δ*F/F* after-before: −0.19 ± 0.03); these events were also slower to reach the peak (Δ*t* from rear to 10% rise: 0.45 ± 0.09 s) and shorter in duration

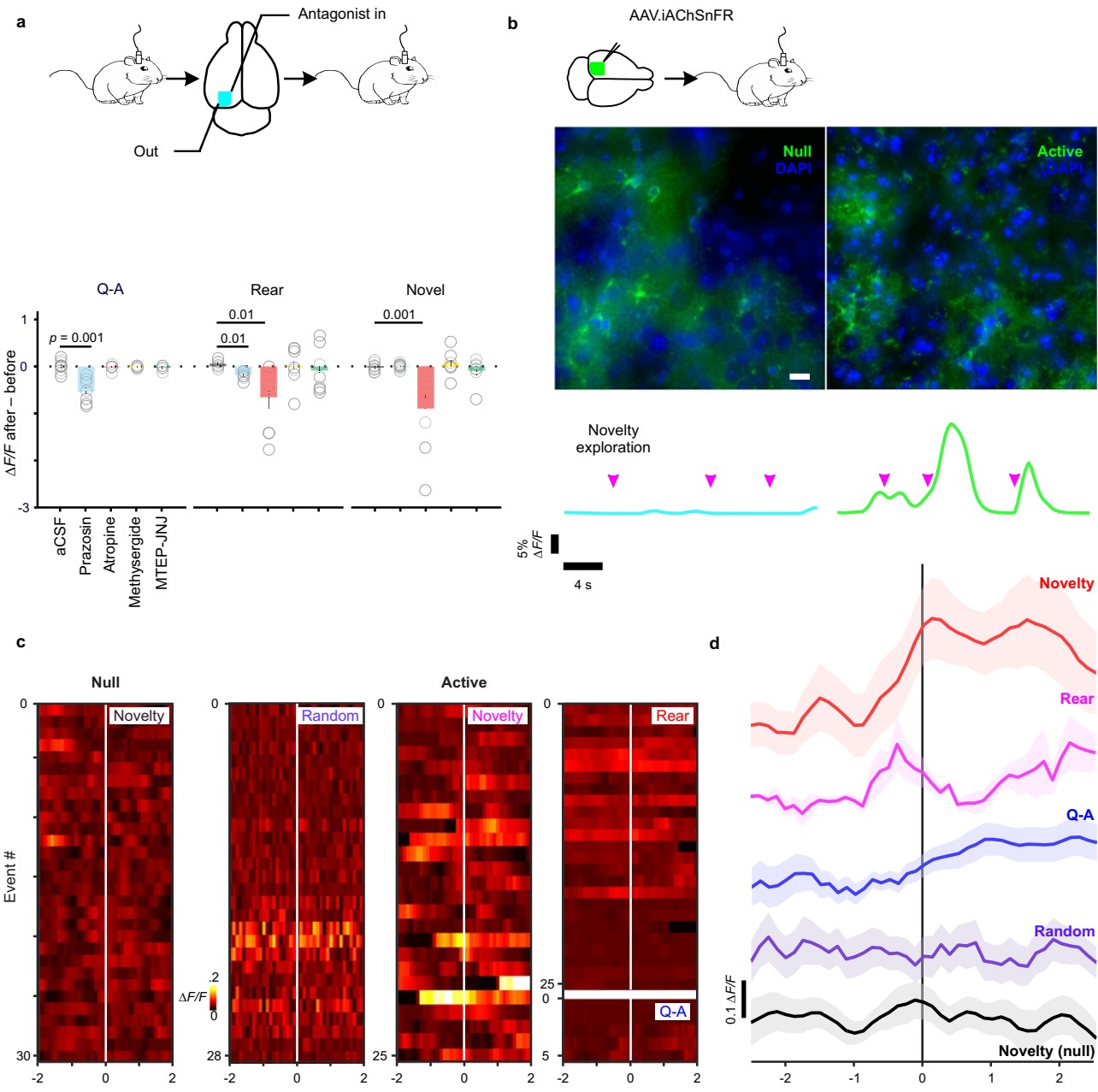

**Fig. 7 | Divergent contextual encoding in cortical astrocytes depends on convergent neuromodulation. a** Top: scheme of topical pharmacology in freely behaving animals. Bottom bar chart: prazosin (eight animals) altered all aspects of QA-relevant astrocyte response and diversive spatial exploration; atropine (nine animals) had a profound effect on amplitudes in contexts of exploration, be it diversive or inspective (Kruskal–Wallis with post hoc Dunn's multiple comparison, global $p = 0.001$). **b** Top: scheme of stereotactic introduction of *AAV.GFAP.iAChSnFR* $^{Active}$ and *AAV.GFAP.iAChSnFR* $^{Null}$ into the mouse cortex. Middle: immunofluorescent images of cortical astrocytes expressing *iAChSnFR* $^{Null}$ (Null) and *iAChSnFR* $^{Active}$ (Active); the preservation of fine ramification indicated no overt

pathological changes in transduced astrocytes. Scale bar is 50 μm. Bottom: representative traces showing directed exploration (magenta arrow head) reliably elicited fluorescence transients in the brains of iAChSnFR $^{Active}$ animals but not in those of the iAChSnFR $^{Null}$ ones. **c** Color raster of fluorescence fluctuations aligned to individual events: directed exploration (Novelty in black) in the iAChSnFR $^{Null}$ animal and in the behaving iAChSnFR $^{Active}$ animal, randomly selected segments (Random), diversive exploration (Rear) and arousal (Q-A). **d** Average across animals for each condition above (12 iAChSnFR $^{Null}$ and 11 iAChSnFR $^{Active}$ animals, mixed-effect model, global $p < 0.0001$). Data are presented as mean ± s.e.m. Source data are provided as a Source Data file.

(FWHM: $-0.15 \pm 0.45$ s, $n = 9$) as compared to aCSF control ($-0.08 \pm 0.08$ s, $0.06 \pm 0.03$ $\Delta F/F$, $-0.09 \pm 0.35$ s, $n = 8$) (one-way ANOVA for $\Delta t$ $_{from\ rear\ to\ 10\%\ rise}$, $F(4, 38) = 5.5$, $p = 0.01$; Kruskal–Wallis for $\Delta F/F$ $_{after-before}$, $H = 12.7$, $p = 0.01$; one-way ANOVA for FWHM, $F(4, 38) = 3.4$, $p = 0.02$) (Fig. 7a, Rear). Together, these results indicate that α1 adrenoceptors play a critical role in engaging astrocyte networks in behaviorally relevant contexts associated with increases in arousal/ vigilance.

Previous studies have shown that electrical stimulation of nucleus basalis or direct application of ACh in vivo can elicit calcium increases in cortical astrocytes[65,66], and single-cell RNAseq analysis of astrocytes in the visual cortex indicate that they express muscarinic receptor M1 (*Chrm1*)[17,25], but the physiological contexts under which these receptors are activated is unknown. To test whether muscarinic receptors contribute to these astrocyte calcium changes, we locally applied the muscarinic antagonist atropine and measured their responses during

different behavioral sequences. Atropine significantly reduced calcium transient amplitudes during both diversive exploration ($-0.66 \pm 0.23$ $\Delta F/F$, $n = 9$ vs. aCSF: $0.06 \pm 0.03$, $n = 8$) (Kruskal–Wallis, $H = 12.7$, $p = 0.01$) (Fig. 7a, Rear) and inspective exploration ($-0.90 \pm 0.27$ $\Delta F/F$, $n = 9$ vs. aCSF: $-0.02 \pm 0.03$, $n = 11$) (Kruskal–Wallis, $H = 18.1$, $p = 0.001$) (Fig. 7a, Novel). In contrast, local inhibition of 5-HT$_{1/2}$ (Methysergide) or mGluR$_{1/5}$ (MTEP-JNJ) signaling did not significantly alter the characteristics of astrocyte responses. However, during inspective exploration, inhibiting mGluR$_{1/5}$ enhanced the synchrony of astrocyte activation (MTEP-JNJ SI $_{after-before}$: $0.06 \pm 0.02$), an effect opposite to that of α1 adrenergic antagonism (prazosin SI $_{after-before}$: $-0.10 \pm 0.04$) (one-way ANOVA, $F (4, 42) = 2.9$, $p = 0.03$). These findings suggest that cholinergic signaling in astrocytes is engaged during particular exploratory events, and that the coordination of astrocyte calcium transients can be altered by signaling through metabotropic glutamate receptors.

As an alternative method to explore whether astrocytes are exposed to ACh released from cholinergic terminals in specific behavioral contexts, we expressed a fluorescent ACh sensor selectively on the surface of astrocytes and monitored fluorescence changes during novelty exploration. The genetically encoded ACh sensor iAChSnFR[67] was expressed on cortical astrocytes by stereotactic delivery of *AAV.GFAP.iAChSnFR* $^{Active}$ into V1 (see "Methods"); control mice were injected with the non-functional variant *iAChSnFR* $^{Null}$ using the same approach (Fig. 7b, scheme). Viral delivery of these transgenes resulted in sparse, but specific, expression of the null and active sensors on astrocytes (Fig. 7b, images). For mice engaged in novel object-directed exploration, each inspection event (Fig. 7b, magenta arrowheads) elicited fluorescence changes in astrocytes expressing iAChSnFR $^{Active}$, but not iAChSnFR $^{Null}$ (Fig. 7b, traces). We also recorded the fluorescence changes from these two proteins when animals explored environmental novelty and transitioned from quiescence to activeness. The corresponding fluctuations were aligned to each directed exploration (Fig. 7c, color raster of Novelty under Null and Active), to each diversive exploration (Fig. 7c, color raster of Rear under Active), and to each Q-A transition (Fig. 7c, color raster of Q-A under Active). For control purposes, a color raster is also presented for randomly-sampled time segments (Fig. 7c, color raster of Random under Active). These changes were then summarized across animals (Fig. 7d, iAChSnFR$^{Active}$, $n = 11$ vs. iAChSnFR$^{Null}$, $n = 12$). This analysis revealed that marked fluorescence increases were observed during each of these behaviors in animals expressing iAChSnFR$^{Active}$, but not those with the non-functional iAChSnFR$^{Null}$ (Fig. 7d, $\Delta F/F$ $_{Active - Null} = 0.31 \pm 0.02$, 95% CI, [0.29, 0.33]) (mixed-effect model with *post hoc* correction, $t (60) = 13.34$, $p = 0.02$). No significant fluorescence changes were detected in randomly selected time segments (Fig. 7d, $\Delta F/F$ $_{random} = 0.00 \pm 0.01$, 95% CI, [−0.02, 0.02]). The ACh signals induced were distinct between Q-A transitions ($\Delta F/F$ $_{Q-A, 500 ms} = 0.09 \pm 0.02$, 95% CI, [0.06, 0.13]), rearing explorations ($\Delta F/F$ $_{Rear, 500 ms} = 0.03 \pm 0.01$, 95% CI, [0.01, 0.05]) and novelty inspections ($\Delta F/F$ $_{Novelty, 500 ms} = 0.14 \pm 0.02$, 95% CI, [0.10, 0.19]) (Fig. 7d); these differences were also time-dependent (mixed-effect model, $F (1.66, 66.3) = 221$, $p < 0.0001$). These findings indicate that ACh released in the cortex during novelty exploration[68,69] reaches astrocyte membranes, providing a means to modulate their physiology.

### Direct modulation of astrocyte signaling by internal state

Astrocytes exhibited few calcium transients when engaged in maintenance behaviors, such as drinking, eating, grooming and nesting under ad libitum conditions (Fig. 4c). However, these fundamental behaviors are also embedded with a motivational component, where the rewarding properties of resources can be magnified through deprivation[70,71]. To examine the effects of internal homeostatic state on astrocyte responses to nutritional cues, we imaged the same mice across water- or food-restricted vs. naive states (scheme, Fig. 8a). Consistent with observations made during home cage monitoring

(Fig. 4c), astrocyte activation was not detected in satiated animals (water: $\Delta F/F$ $-0.12 \pm 0.05$ in 24 mice, food chow: $\Delta F/F$ $-0.19 \pm 0.08$ in 20 mice) (Fig. 8a). Strikingly, in animals subjected to water deprivation ($\Delta F/F$ $0.47 \pm 0.12$ in 17 mice) or food deprivation ($\Delta F/F$ $0.46 \pm 0.21$ in 22 mice), astrocytes were robustly activated when mice were exposed to these previously neutral cues (Fig. 8a). However, this increase was only significant in water-deprived conditions, due to high variability among animals (difference of $\Delta F/F$ in water-deprived vs. satiated animals: 0.21, [0.01, 0.41]; in food-deprived vs. satiated: 0.18, [−0.02, 0.39]) (water: Wilcoxon rank-sum test, $U = 128$, $p = 0.04$; food chow: Wilcoxon rank-sum test, $U = 154$, $p = 0.10$) (Fig. 8c). Examination of the response profiles of astrocytes revealed that their initial response occurred prior to drinking (calcium peak time: −1.5 s) or ingesting food (−1.75 s). Thus, we hypothesized that these deprivation-induced changes were triggered primarily by cue detection rather than consumption. To test this hypothesis, mice were presented with inaccessible cues (scheme, Fig. 8b) in which they could see or smell food, but not engage with it. Simply detecting the cues without direct interaction resulted in rapid activation of astrocytes (smell: $\Delta F/F$ $2.3 \pm 0.7$ in 22 mice; see: $\Delta F/F$ $1.6 \pm 0.5$ in 25 mice) (Fig. 8b). Here, the difference between the food-deprived and the satiated were much more striking ($\Delta F/F$ $_{deprived - satiated}$ in smelling inaccessible food: 0.80, [0.33, 1.41], Wilcoxon rank-sum test, $U = 51$, $p < 0.0001$; $\Delta F/F$ $_{deprived - satiated}$ in seeing: 0.67, [0.41, 1.18], Wilcoxon rank-sum, $U = 66$, $p < 0.0001$) (Fig. 8c). Of note, these responses were much larger in magnitude than when they were allowed to eat (Kruskal–Wallis, $H = 16.3$, $p = 0.0002$), suggesting that the inability to obtain the goal may unmask and/or provoke greater neuromodulatory release as a consequence of enhanced and sustained awareness. Together, these findings indicate that the engagement of cortical astrocytes is not fixed within different behavioral contexts, but instead can be adjusted based on changes in internal motivational state.

## Discussion

Astrocytes are now recognized as both a target of neuromodulatory influence and an effector that can induce broad changes in neural activity during different behavioral states[4,6,8,9], but the relationship of their activity patterns to diverse naturalistic behaviors remains to be fully defined. The increasing recognition that cortical astrocytes vary in their molecular and physiological properties, between brain regions[72], cortical layers[16,17], and within domains of particular signaling such as sonic hedgehog[2,15], necessitates cellular rather than population measures of activity. To expand our understanding of the relationship between astrocyte calcium dynamics and animal behavior, we optimized a high-resolution, long working distance, multicore fiber optic imaging system capable of resolving GCaMP fluorescence changes induced by fluctuations in intracellular calcium within cortical astrocytes in freely behaving mice. By implementing this imaging modality in different behavioral contexts, we show that cortical astrocytes exhibit remarkably diverse calcium activity patterns, which are modulated by internal states and exhibit features appropriate for integrating state transitions over the sleep-wake cycle. This expanded knowledge of the astrocyte behavior-activity framework will help to define the molecular mechanisms that control astrocyte activity and the consequences of distinct activity patterns on neuronal activity.

Multicore optical fiber bundles are frequently used to deliver light to activate optogenetic effectors and monitor fluorescence changes using photometry[40,73]. In detection mode, such systems necessarily integrate light from various sources, providing an ensemble average of population activity dictated by the expression of the sensor, the scattering properties of the tissue, and the photon capture capabilities of the optical path. Coherent multicore fiber bundles use many thousands of optically isolated fibers to channel light to the detector, preserving spatial information. Here, we used an objective lens assembled from two pairs of aspheric miniature lenses to collect and

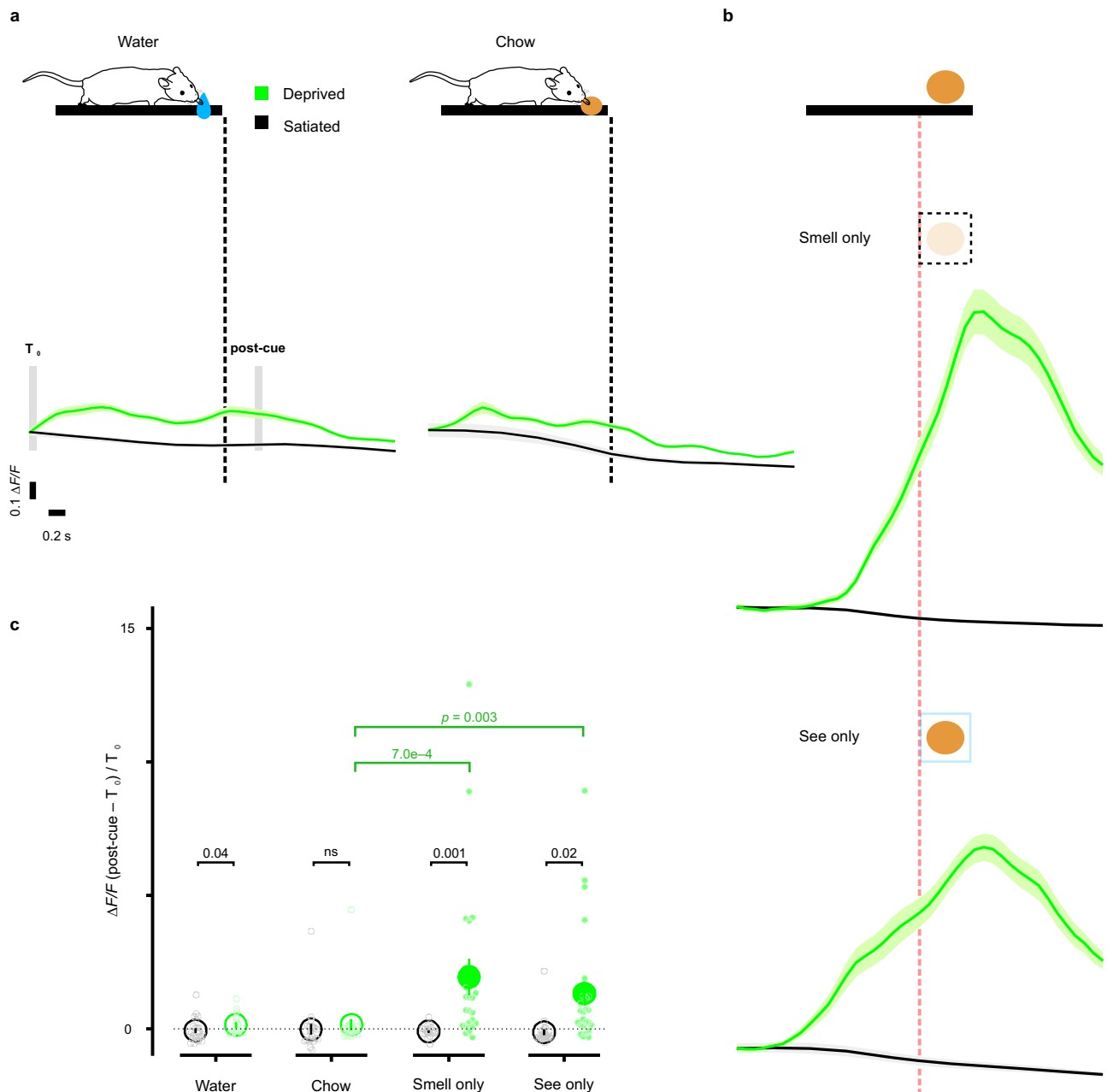

**Fig. 8 | Internal state-driven astrocyte signaling of nutritional cues. a** Imaging the same animal across water- (left) or food-deprived (right) vs. satiated states; astrocyte activation was not seen in metabolically-satiated animals (water: $\Delta F/F$ (post-cue 200 ms − T0)/T0 −0.12 ± 0.05, food: $\Delta F/F$ (post-cue 200 ms − T0)/T0 −0.19 ± 0.08). The population became sensitive to cues during behavioral states of thirst ($\Delta F/F$ (post-cue 200 ms − T0)/T0 0.47 ± 0.12) or hunger ($\Delta F/F$ (post-cue 200 ms − T0)/T0 0.46 ± 0.21). **b** Inaccessible cues augmented astrocyte activation (smell: $\Delta F/F$ (post-cue 200 ms − T0)/T0 2.3 ± 0.7; see: $\Delta F/F$ (post-cue 200 ms − T0)/T0 1.6 ± 0.5) compared to those of the accessible, a phenomenon that was not seen in satiated animals (satiated-smell: $\Delta F/F$ (post-cue 200 ms − T0)/T0 −0.02 ± 0.01; satiated-see: $\Delta F/F$ (post-cue 200 ms − T0)/T0 0.01 ± 0.01) (Wilcoxon rank-sum test, global $p < 0.0001$). **c** Consistent astrocyte activation to cues in nutritionally-deprived animals (22 mice) across accessible and inaccessible cues (black asterisks) and augmented responses to inaccessible cues (green asterisks, smell- or see-only vs. eat) (Kruskal−Wallis with *post hoc* Dunn's multiple comparison, global $p = 0.0003$). Data are presented as mean ± s.e.m. Source data are provided as a Source Data file.

focus emitted light onto the end of a 30,000-core fiber bundle to enable cellular resolution of active regions[40,42]. Importantly, by using lenses with an extended focal length, this removable, lightweight, small form factor device enabled longitudinal imaging through an implanted cranial window with minimal behavioral interruption[45], providing sufficient spatiotemporal resolution and sensitivity to resolve cellular fluorescence changes within the intact brain while animals were engaged in normative behaviors. Although this system operates in an epifluorescence mode, each optical fiber core intrinsically acts as a spatial filter to prevent photons from outside the focal plane to infiltrate, providing better optical sectioning than a traditional wide-field epifluorescence microscope[33,74]. The ability to modify imaging parameters, such as light sources, filters, and optic sensors remotely using standard tabletop optical components in this system affords rapid and extensive adaptation to experimental needs. Further modifications to this system could be implemented to enable deeper structure observation and high resolution with multi-photon excitation and/or adaptive optics; fast three-dimensional imaging using adaptive lenses and laser scanning; multicolor detection and laser speckle contrast imaging to resolve interactions between cell types

and blood flow; and holographic or random-access fiber illumination to manipulate cellular activity patterns using optogenetics[33,45,75,76]. Although this system requires fewer miniature elements than head-mounted "miniscopes"[77], it requires delicate assembly of the multi-lens array and fiber optic coupling[40,42]. Additionally, although animals acted similarly when tethered or untethered, the multicore fiber is more rigid than electrical wires used for data transfer in miniscopes, which could potentially influence more subtle behaviors.

Among other behaviorally relevant state changes, astrocytes have been implicated in controlling sleep homeostasis in multiple species[29,37,51]. Astrocyte calcium levels are thought to be the primary means through which these cells accumulate information about the circadian cycle, a hypothesis supported by pioneering imaging studies in unrestrained mice using fiber photometry and head-mounted miniscopes[26,34]. By implementing this multicore optical fiber imaging system, we were able to extend this analysis over 24 h imaging periods to resolve longitudinal activity patterns of individual astrocytes within the cerebral cortex, providing direct evidence that calcium activity within individual astrocytes varies prominently with the internal clock of the animals. These findings are consistent with findings that astrocytes, in addition to encoding episodic sleep-wake state transitions[26,34,78], can also store an extended record of sleep required for sleep homeostasis (Supplementary Fig. 5)[12,29,52], perhaps through activation of calcium-dependent changes in gene expression[32,79,80].

Using simultaneous astrocyte calcium imaging and behavioral tracking, we linked astrocyte calcium changes to other distinct behaviors using a semi-automated behavior classifier (Fig. 4). Consistent with studies in head-immobilized animals[40,77], we observed that Q-A transitions were associated with global, transient activation of cortical astrocytes. In contrast, there was a difference among self-paced behaviors, specifically in exploration vs. maintenance behaviors, with only the former associated with a calcium rise. This differential astrocyte response to behaviors that share similar engagement of motor programs argues against ongoing movement/motor/muscular information processing by cortical astrocytes, a conclusion also drawn from observing animals performing Q-A transitions[81]. It is notable that, during maintenance behaviors, astrocyte activity fell below the pre-behavioral value (Figs. 4c and 8), suggesting that there may be bidirectional modulation of cortical astrocytes, either from active suppression[82] or from a decrease in neuromodulatory tone, perhaps reflecting a general decrease in noradrenergic or cholinergic neuron firing during satiety[77].

To explore with which behavioral features astrocyte activities are most closely associated, we adopted paradigms where mice were allowed to explore arenas and investigate novel objects. These sets of experiments provided behavioral distinctions between exploration and non-exploration. Notably, in paradigms of diversive exploration, it was the behavioral termination that dominated ensemble astrocyte calcium elevations (Fig. 5)[83]. Offset-responsive astrocytes might therefore reflect internal adjustments, such as deviations of expected input from contextual feedback. This feature is of particular relevance for generating appropriate behavioral responses to deviations from expected information. Consistent with this conclusion, primary cortices in rodents have been shown to not only represent primitive inputs/outputs, but also act as feedback detectors, essential for rapid initiation of behavioral adaption to the unexpected[23,84]. Indeed, recent studies indicate that new inferences of uncertainty are made based on local cholinergic and noradrenergic suppression of intracortical transmission[24].

Directed exploration paradigms provide unambiguous discrimination of investigatory behaviors and provide an opportunity to observe specific responses of individual cortical astrocytes (Fig. 6). Consistent patterns of cortical astrocyte activity across object interactions were reminiscent of the attractor network underlying internal states[85]. Within a given group of astrocytes, the differential timing of

single cells in sub-clusters suggests context-dependent modulation of individual cells. We conclude that, together with known changes of ensemble activity that serve as a basic structure, the staggered recruitment of astrocytes could prolong neuromodulatory events and provide a means to encode distinct temporal sequences of neuronal activity.

Intriguingly, during exploration directed to novel objects, astrocytes exhibited a trial-by-trial reduction in calcium transient magnitude. This fast adaptation implies that the responsiveness of cortical astrocytes, on a population level, was updated rapidly when animals were in a familiar context. By suppressing the responses to experienced contexts, cortical astrocytes may gain the capacity to encode contingencies more efficiently and to maintain responsiveness to rare saliencies[22]. However, within a population there were always astrocytes that did not exhibit the principal features of defined clusters. These cells were either unresponsive throughout or presented unreliable patterns across trials. As in the case of fast-adapting ensembles, cortical astrocytes that fell out of clusters in this novel object-exploring context could serve as a reserve to permit circuits to encode a broad range of salient features[86].

To investigate the molecular mechanisms that enable engagement of cortical astrocytes in different contexts (Fig. 7), we performed local pharmacological manipulations to restrict receptor engagement to superficial cortical areas. As shown previously, NE induced consistent increases in calcium throughout the imaging field during Q-A transitions through activation of α1 adrenoceptors[27,40]. Moreover, NE and ACh inputs converged and enabled synergistic activation during diversive exploration. Mechanistically, once receiving convergent or divergent inputs, individual astrocytes may interpret and integrate various time-dependencies via crosstalk between G-protein-coupled receptors. For example, Gαi/o- and Gαq-linked G-protein-coupled receptors have been shown to yield synergistic calcium responses in astrocytes of young animals[87]. Such interactions could allow more diverse outcomes, as recent studies indicate that chemogenetic activation of Gαi/o or Gαq pathways in astrocytes induced distinct changes in non-rapid eye movement, sleep depth and duration[88].

In contrast to the NE-dominant response during Q-A transitions, ACh influenced the amplitude of astrocyte calcium transients during diversive explorations, while NE shaped the time course and timing relative to the behavioral sequence. This convergent action may reflect the ambiguous state change involved not only in exploring, but also in confronting the unexpected. Consistent with this possibility, when animals directed their exploration to an unambiguous target, mAChR signaling became more prominent. Using a new genetically encoded optical sensor for ACh (iAChSnFR) with hundred-millisecond kinetics, we found that this ACh reaches astrocyte membranes when mice explored novel objects[67]. A direct action of ACh on astrocytes is also supported by the results of in vivo topically applied antagonists in the context of minimal endogenous neural and synaptic activity[30,66]. Our studies also suggest that α1 adrenoceptors and group I mGluR signaling have contrasting effects on cell synchrony, with the former enhancing astrocyte network synchrony and the latter promoting desynchrony. This glutamatergic network desynchronization implies that modulation of astrocytes by local neuronal activity or primary sensory inputs, while subtle, does exist[89]. This is significant because direct engagement of these inputs in vivo does not, by itself, promote the widespread activation of astrocytes comparable to that induced by NE during periods of heightened arousal[27,40,77]. Thus, different neuromodulators released at specific times during a behavioral sequence appear to converge on astrocytes to manipulate distinct aspects of their calcium activity, suggesting that this is a highly nuanced signaling mode that varies both in space and time depending on which neuromodulators are present.

Many behaviors are strongly modified by internal state. We also explored whether changes in motivation alter the response of

astrocytes, focusing on maintenance behaviors (drinking, feeding), which were shown to elicit no activation in cells in animals provided unlimited access to food and water. If cortical astrocytes encode salient contexts, manipulating the internal state of animals without changing the external environment should also enforce state-dependent activation of these cells[90,91]. In keeping with this prediction, perturbed homeostasis, like food and water restriction, reversed astrocyte calcium responses from decreasing to profoundly increasing (Fig. 8). We further determined that exposing animals to food, but preventing consumption, initiated responses that were even stronger than if they were allowed to eat; the strongest response was observed in cases where mice were able to smell the resource but could not see or access it. This phenomenon further strengthens the conclusion that cortical astrocyte activity is aligned with contextual saliencies, likely reflecting differences in the level of neuromodulatory engagement when goals are predicted but not achieved.

Despite many advances in modeling neural activity in complex networks, astrocytes have been characteristically excluded, since much less is known about their activity patterns within specific behavioral contexts or the effects of this activity on surrounding cells. Recent studies have attempted to address this question by activating or silencing astrocytes within circuits. However, interpretation of these findings may be complicated by limitations in our ability to mimic the normal spatiotemporal characteristics of their activity or to block these dynamic calcium changes transiently during specific, appropriate behavioral sequences. This challenge may explain why manipulations of astrocytes using standard opto- or chemogenetic approaches, have yielded conflicting results[89,92–94]. New tools are being developed to manipulate calcium signaling in astrocytes, such as the expression of a plasma membrane calcium ATPase to suppress calcium transients[79], melanopsin to induce calcium activity[95], and long-wavelength opsins to allow modulation of astrocytes in deeper brain regions without fiber or lens insertion[96], which may further extend our understanding about the consequences of astrocyte calcium dynamics during different behaviors. The profound consequences of these manipulations raise the intriguing possibility that dysregulation of astrocyte calcium signaling contributes to neuropsychiatric diseases[3,30,72,92]. In vivo analyses, such as those enabled by this fiber optic imaging platform, will help reveal the precise nature of astrocyte activity in cortical circuits and provide a physiological framework necessary to define their contributions to specific behaviors.

There are several limitations of the multicore imaging platform and the results obtained in this study. Under ideal conditions the multicore system can achieve subcellular resolution (Fig. 1); however, the highly scatting nature of brain tissue reduces the resolving power of this epifluorescence system in vivo. For this reason, our findings represent aggregated activity over the processes and soma of cells. Astrocytes exhibit highly localized microdomain calcium transients in their processes that are produced through mechanisms distinct from activity in the soma[28,97]. Further studies using membrane anchored sensors, which provide better detection of calcium changes within astrocyte processes[11,97], or other methods to restrict localization, could reveal compartment-specific changes in astrocyte calcium that encode specific aspects of these complex behaviors[11,18,97]. Photon scattering also limits the depth of imaging to the upper layers of the cortex. Implantation of the device would allow detection of astrocyte activity in other cortical layers and deeper brain structures; however, our goal was to design a system that could be placed outside the brain with sufficient working distance to image through an implanted cranial window, so as to limit the extent of tissue damage. We optimized illumination parameters to minimize both photobleaching (less than 0.1% fluorescence decay over 24 h) and phototoxicity (no increases in GFAP or additional lipofuscin accumulation observed following imaging), but it is possible that surgical manipulation, imaging, and tethering of the mice, could have influenced the responses of

astrocytes and the behaviors of the mice. We employed local application of pharmacological agents to explore the involvement of different neurotransmitters and receptors in astrocyte calcium signaling. However, because the receptors targeted by these compounds are expressed by both neurons and glia, we cannot attribute changes in astrocyte activity during these experiments to the direct inhibition of astrocytic receptors[17,20,25]. The neurotransmitter profiles present during these different behaviorally induced forms of astrocyte calcium signaling motivate further mechanistic exploration of possible direct modulation of astrocytes during many physiologically relevant behaviors.

## Methods
All experiments and procedures were approved by the Johns Hopkins Institutional Care and Use Committee. Requests for sharing resources (see Supplementary Table 1) should be directed to the Lead Contact, Dwight Bergles (dbergles@jhmi.edu).

### Experimental model and subject details
**Subject.** All experiments and procedures were conducted in accordance with the Johns Hopkins Institutional Care and Use Committee animal protocol number MO15M308. We included in this paper mice of random sex, mixed *B6N;129* background (from genetically-modified animal crosses as below) and between 20 and 24-weeks-old. The animals could assess water and food ad libitum in standard polycarbonate cages with enrichment. The housing facilities are maintained at 40–60% humidity, at a temperature of 20–25 °C and on a 12-h light/dark cycle.

**Transgenic models.** Generation of the following mouse lines have been previously published: *Tg(Slc1a3-cre/ERT)1Nat/J*, also known as *GLAST-CreER*; STOCK *Gt(ROSA)26Sor^{tm1.1(CAG-EGFP)Fsh}/Mmjax*, also known as *RCE:loxP*, *B6N;129-Gt(ROSA)26Sor^{tm1(CAG-GCaMP3)Dbe}/J*, also known as *Rosa26-lsl-GCaMP3*[40]. Corresponding RRIDs are listed in key resources table. We acquired these mice from the Jackson Laboratory and crossed the Cre recombinase-conditional EGFP (*RCE:loxP*) or GCaMP (*Rosa26-lsl-GCaMP3*) to the Cre-bearing (*GLAST-CreER*) mice. Offspring (*GLAST-CreER; RCE:loxP* or *GLAST-CreER; Rosa26-lsl-GCaMP3* mice) were then exposed to tamoxifen to induce selective expression of EGFP or GCaMP in astrocytes.

### Experimental details
**Tamoxifen injections.** The transgenic mice received three intraperitoneal injections of 100 mg/kg body weight tamoxifen (T5648-1G, Sigma-Aldrich, CAS: 10540-29-1), dissolved in sunflower seed oil (S5007, Sigma-Aldrich, CAS: 8001-21-6), within 5 days during the fourth postnatal week.

**Cranial window implantation.** The 20-week-old mice were anesthetized using isoflurane (Baxter International) flowed in $O_2$ (4% in 1 l/min for induction and 1.5% in 0.5 l/min for maintenance, targeting at 1–2 breaths per second). The central temperature was kept at 37.5 °C with a feedback controller (TC-1000, CWE Inc). We adopted peri-operative occlusion and lubricating eye-ointments (GenTeal PM) to prevent ocular complications. After thorough shaving and skin asepsis using three alternative swabs of 70% Ethanol & 10% Povidone-Iodine, a 10 mm-long rostro-caudal opening, revealing landmark sutures, was made in the scalp. Then, subcutaneous tissue and muscles were trimmed and periosteum removed. Tissue margins were secured with super glue (KG 483, Krazy Glue). A stainless steel headplate with a central opening was cemented to the exposed skull (C&B Metabond, Parkell; 1520 BLK, Lang Dental). Animals were then positioned to a head-fixed apparatus. Using a hand-piece micro-drill (XL-230, Osada) drilling beyond the cancellous bone, followed by custom ophthalmic blade (Superior Platinum, ASTRA; 14134, World Precision Instrument)

cutting the inner compact bone, a craniotomy (2.8 mm × 3.0 mm) was made at 1.0 mm anterior to lambda and 3.1 mm lateral from midline. A custom single-edge rounded trapezoidal glass coverslip (85–130 μm in thickness) was thereafter placed and cemented in place (C&B Metabond, Parkell; 1520 BLK, Lang Dental).

**In vivo imaging.** A 0.75 W, 473 nm, 0.8 mm beam of an optically pumped semiconductor laser (OBIS473LX, Coherent) was expanded (Olympus PLN 20×/0.4) and reflected by a dichroic mirror (FF499-Di01-25×36, Semrock). The expanded and reflected beam was then coupled into a multicore optical fiber bundle (FIGH-30-650S, Fujikura) via a focusing objective (Olympus PLN 10×/0.25). At the distal fiber end, light was delivered to tissue through a 470–570 nm achromatic pair of aspheric doublets (352140-A, Thorlabs) housed in stainless steel tubes (Small Parts). The laser power under the fiber tip was 0.7–0.9 mW. Returning light travelled back via the lens probe, the optical fiber, the coupling objective, the dichroic mirror, an emission filter (MF525-39, Thorlabs), a fixed lens (AC254-150-ML-A, Thorlabs) and was focused onto a CCD (GS2-FW-14S5M-C, FLIR).

For the reflectance imaging setup, dual-spectral reflectance at major excitation & emission wavelengths of GCaMP fluorescence imaging were sampled as reported[49]. Using a long-pass dichroic (FF499-Di01-25×36, Semrock), we combined cyan (OBIS473LX - measured peak $\lambda$: 473 nm -, Coherent) and green (PLP520 - measured peak $\lambda$: 523 nm -, Osram) illumination into one path, further coupled into the fiberscope described in the section above. Using a beamsplitter (EBS1, Thorlabs), reflectance signals were directed into the detecting path, and images were acquired with a single 16-bit depth scientific complementary metal-oxide-semiconductor camera (C11440-42U30, Hamamatsu Photonics).

We exploited a motorized micromanipulator (MX7630L, DR1000, MC1000e, Siskiyou; GNL10, Thorlabs; custom machine parts) for precision positioning of the fiberscope to a desired FOV; mini-objective mounts were nutted down on headplates to allow optics-animal coupling; FOV was then secured using screws to hold the mini-objective in place (Fig. 2a). Data were acquired at 20 frames per second, on-line averaged to 2 or 4 frames per second, with a resolution of 1280 × 960 pixels using commercial software development kit (FlyCapture SDK, FLIR) integrated into the master program (C#, Microsoft).

**Quantitative assessment of behaviors.** The imaging suite was set up in an isolated room so as to limit potential stressors. Ceiling-reflected incandescent lights at corners of the room provided lighting measured 28–32 lux (0.04 W/m², 555 nm) in the center of mice behaving area. Ambient sound was masked by a white noise machine (BRRC112, Big Red Rooster). After recovering from window implantation, mice bore weights on their heads (3.0 g) in cages for two weeks; during the same period of time, we habituated mice to experimental settings. Experiments were carried out in the fourth week, and each animal would engage in only one single paradigm unless otherwise mentioned. Mice lived on a 12-h light/dark cycle, and were subjected to experiment during their dark, active phase (between 18:00 and 06:00), except for an independent group of animals who underwent 24 h imaging across their circadian cycle. We transported mice to the suite 1 h prior to testing, and during experiment, stayed the unnoticeable to them.

Two near infra-red cameras (NIRCam) (DCC1645C, Thorlabs—with IR filter removed, or FI8910W, Foscam) were centered 100 cm above (NIRCam-1), and on the same level 25 cm (NIRCam-2) from the experimental platform. Videos were recorded at 30 frames per second with a resolution of 1280 × 960 pixels through master program capture of video streams (Foscam VMS, Foscam; C#, Microsoft).

For the head-fixed quiet awake to active awake (Q-A) paradigm, mice were placed on a custom head-fixed rotating platform (S1-Round12-.125, Source One), set on the stepper motor-driven mode

(ROB-09238, ROB-12779, SparkFun Electronics). Rotation was monitored with an optical encoder (600128CBL, Honeywell). An Arduino Mega 2560 R3 microcontroller board (191, Adafruit) signaled motor onsets and digitized encoder readings.

To assess the influence of fiberscope-tethering on animals, mice explored a 30 × 30 × 30 cm³ chamber for 30 min. Then we recorded locomotion for the following 30 min. A second group of mice went through the same protocol but without being tethered to the fiberscope. Before testing and between mice, the chamber was cleaned with three alternative washes of purified $H_2O$ and 70% ethanol.

For longitudinal behavioral cycle imaging, mice were habituated to the imaging chamber for increasing durations beginning two weeks prior to the imaging day. In the 3 days prior to the imaging day, mice were tethered to the fiberscope and allowed to freely move for 60 min each day. On imaging day, mice were tethered and freely moving in the imaging cage for 60 min prior to the start of data collection. Mice were then allowed to explore the chamber, with the experimenter monitoring the mice for stress or discomfort remotely. Recordings were terminated after 26 h, after which mice were untethered and returned to their home cage. To avoid potential confounds from handling and stress, the first 2 h of data collection were discarded.

In order to phenotype spontaneous behavior of mice in home cages, we separated and housed individual mice in single cages 26–28 h prior to imaging day. On the day of experiment, we performed imaging when each mouse behaved freely in its home cage. During imaging, an unfamiliar object to the mouse was placed in the cage to stimulate the exploratory and investigatory aspects of behaviors, which animal would infrequently do at home. After experiments, mice were re-introduced with their previous cage mates under supervision.

To observe diverse exploration in environmental novelty, mice were placed in the center of a 30 × 30 × 30 cm³ chamber and imaged for 5 min. Upon each mouse completing a trial, we placed it into a new, interim cage until all his or her cage mates have been tested. Then mice from the same original cage returned to their home. A separate group of mice was allowed to explore the chamber for 60 min 24 h before. They were returned to the home cage afterwards, and then participated in the same paradigm as the experimental animals on the day of imaging. Before testing and between mice, we cleaned the chamber with three alternative washes of purified $H_2O$ and 70% Ethanol.

For inspective novelty exploration, on the day before imaging, mice got habituated to a 40 × 20 × 30 cm³ chamber for 60 min. On imaging day, mice first interacted and became familiar with two identical objects (positioned at two interior corners of chamber, 5 cm from walls) for 10 min. Afterwards mice stayed in an interim cage, during which we replaced the two objects, one with an identical triplicate and the other with a novel object. Within an hour, mice returned to the testing chamber and explored the familiar vs. the novel object for 10 min at free will. We used 250 ml polypropylene copolymer jars (about 6.4 cm in diameter and 11.5 cm in height) of light blue desiccants (2118-0008, Thermo Fisher Nalgene; 23001, Drierite), and towers of polyethylene-wrapped colored polyhedron sitting on color-taped optical posts (RS3P4M, Thorlab) (about 5.0 cm in diameter and 14.0 cm in height) as the target objects. Choice of objects for familiar and novel trials were based on a crossover design to avoid confounding due to differences in objects. Before testing, between sessions and between mice, we cleaned the chamber with three alternative washes of purified $H_2O$ and 70% Ethanol.

Laser illumination, brain imaging and behavior recording were synchronized on a generic data acquisition board (USB 6009, National Instruments). When behavioral recording cannot be triggered on the master clock, the frame clocks of the CCD were duplicated to set on a 940 nm light emitting diode (H&PC-56931, Chanzon), recorded in NIRCam recordings as timestamps; local time at millisecond precision was also tagged in NIRCam images (DateTime.Now.ToString; C#). We programed the master control on the

basis of off-the-shelf development kits (FlyCapture SDK, FLIR; Foscam VMS, Foscam; Measurement Studio Standard, National Instruments; C#, Microsoft) running in a standard x64 operating system (Windows 7 Enterprise, Microsoft), on a custom-assembled desktop computer (MZ-7KE256BW, Samsung; BX80648I75930K, Intel; CMK32GX4M2B3200C16, CORSAIR; STRIX-GTX960-DC2OC-2GD5, Asus; GA-X99-UD3P, GIGABYTE).

**Pharmacology.** We performed systemic pharmacology by injecting mice who participated in freely moving imaging intraperitoneally with 10 ml/kg 0.9% NaCl (vehicle) in control mice or 10 ml/kg 0.9% NaCl with dexmedetomidine (3 μg/kg) in experimental groups.

For topical pharmacology in freely moving mice, we first fabricated chemical-infusing cannula using methods as previously reported[98]. In brief, a guide cannula of 3.0 mm in length was made out of a 23 G hypodermic needle (305120, Becton Dickinson). We layered epoxy (20945, Devcon) to a half-circle baseplate—about 1.5 mm in radius—at 1.0 mm from one end, denoted as the intracranial end. Second, a 5.5 mm-long stylet was made from a 30 G stainless steel wire (AW1-30-SS, Artistic Wire), inserted and protruding from the intracranial end for 0.1 mm and for 2.4 mm from the other.

Implantation of cannula is enabled in a two-step operation. We first cemented headplates to the skulls of 20-week-old animals, and then delayed the next step of surgery, including implanting windows and cannulas, to the day before imaging in the 24th postnatal week. To provide infusion access to imaging area, two quarter-circles at two diagonally opposite corners—0.35 mm in radius—were cut off from the custom-shaped coverslip. During surgeries, additional skull openings of 0.7 mm in diameter were made at corners of craniotomy matching the quarter-circle cuts of coverslip. Dura was removed. Once coverslip is cemented, we disinfected and filled cannulas with 3 alternative washes of sterile 0.9% NaCl (Baxter Healthcare) & 70% Ethanol, placed stylet-in-guides at two corners, cement-set the baseplate on skull (Grip Cement, Dentsply International), sealed coverslip corners with tissue adhesive (VetBond, 3 M) or UV curable optical adhesive (NOA61, Norland Products), and built layers of cement around guides (Grip Cement, Dentsply International). The extracranial end was bent at right angle and covered with silicone (Ecoflex 5, Smooth-On).

On the imaging day, an internal cannula of 5.5 mm in length was made out of the 30 G hypodermic needle (305106, Becton Dickinson). We fitted 2.4 mm of internal cannula into a PE10 tube (427401, Becton Dickinson) and sealed with epoxy (20945, Devcon). We disinfected and filled the internal cannula with three alternative washes of 70% ethanol and sterile artificial cerebrospinal fluid (aCSF 298 Osm/kg, pH 7.35 adjusted with NaOH), containing in mM: 137 NaCl, 2.5 KCl, 1 $MgCl_2$, 2 $CaCl_2$, 20 HEPES. For mice presenting desired behaviors in a baseline session, we repositioned them in a head-fix apparatus, removed silicone coverings, replaced their two stylets with internal cannulas, rendering ~0.1 mm extension below the guides, estimated to be 0.05–0.1 mm above the cortical surface, extensively sealed the extracranial cannula system (20945, Devcon), perfused the cortex at a flow rate of 0.5 ml/min for 15 minutes with drugs (100 μM prazosin, 100 μM atropine, 30/30 μM MTEP/JNJ16259685, 30 μM Methysergide) dissolved in aCSF, and drained the overflow through the other cannula. Then we removed the internal cannulas, sealed the openings with epoxy (20945, Devcon), released the animals into the behavior chambers, and started a session to evaluate pharmacological effects on astrocyte dynamics.

**Expression of iAChSnFR.** The 20-week-old *GLAST-CreER; Rosa26-lsl-GCaMP3* tamoxifen-naive mice were anesthetized and prepped as above. Once revealing landmark sutures, we made a burr hole (1.0 mm in diameter) in the skull using a micro-drill (Stoelting) at 1.0 mm anterior to lambda and 3.1 mm lateral from midline. Adeno-associated

virus, *AAV.GFAP.iAChSnFR* $^{Active}$ or *AAV.GFAP.iAChSnFR* $^{Null}$ [67], was injected 0.45 mm ventral from dura surface. Using a Nanoject II (Drummond Scientific), 132 nl of 5.0e + 09 gc AAV was delivered through 10× 13.2 nl injections (pausing 10 s in between) via a beveled glass pipette (tip diameter 15-20 μm). After injections, the pipette was removed in steps of 0.1 mm per min, and upon full removal, the burr hole was sealed with tissue adhesive (VetBond, 3 M). A cranial window would then be placed.

**Calcium imaging analysis.** We processed imaging data using routines established upon published MATLAB pipelines (R2019b, MathWorks). Shifts in imaging fields were corrected using hierarchical framework. Each image series was divided into stable vs. unstable segments based on KLT tracking of investigator-validated SURF points in a reference frame (built-in function "estimateGeometricTransform" & system object "vision.PointTracker"); unstable segments were aligned with intensity-based rigid registration ("imregister"); then segments were recombined to form registered series. We then TV-L1 denoised images (Lourakis implementation "TVL1denoise"), and retrieved region of interests (ROIs) using CNMF-E, assuming diameter of the background ring 1.5 timed that of the largest cell[99].

Thereafter, average pixel fluorescence was acquired for each ROI, which was transformed into relative changes in fluorescence, $\Delta F/F_O$. $F_O$ was acquired by kernel density estimate ("ksdensity") or, for 24 h imaging, median of the fluorescence. A single image series of n frames were downscaled to $n/10$ bins, the vector of minimal fluorescence in each bin was calculated, kernel density distributions were estimated through the vector in sliding windows of 25 frames, and then the baseline was approximated to be the modified Akima interpolation ("interp1") of peak values of the distributions. We identified local maxima ("findpeaks") with $z$ scored height >2.5 as tentative peaks, next confirmed with interactive and iterative fitting ("iPeak"). Reflectance measurements were converted into estimates of scattering changes to account for the magnitude of absorbance contamination in both excitation and emission fluorescence bands, in accordance with Beer-Lambert derived formula[49].

When calculating the posterior probability of mouse activeness during an astrocyte event, we used the following values deducted from our data. The probability of mouse being active was 0.24. The probability of astrocytes having an event was 0.09. For the actual data, the probability of astrocyte having an event when mouse was active was $0.21 \pm 0.03$ and for the random process $0.09 \pm 0.01$. We acquired probability density function with 1000 bootstraps or random generators.

For ratio of Fano factor (rFF) computation, we calculated the relative changes in mean astrocyte response and across-event variance[63]. The fractional changes in FF from epochs of a given behavioral dimension (i.e., moment before rearing, onset of rearing, offset of rearing) to epochs of maximal mean response, taking place during Q-A transitions.

Synchronization index (SI) was calculated by amplitude-adjusted Fourier transform and equal-time correlation analysis, using the 5 s around the behavior of interest as the sampling interval. This procedure was carried out using the MATLAB toolbox: Measures of Analysis of Time Series. The peak transients of individual cells was used to yield the correlation matrices via equal-time correlation. To ensure the robustness, we tested this result against surrogate time transient series. The process of generating these resampled data involved using the actual peak transient data to compute time-dependent rates which went through a Poisson process to generate the surrogate series. Passing the surrogate time series through equal-time correlation function iteratively 100 times yielded the distribution of surrogate eigenvalues. We considered a result of experimental correlation as statistically significant when it deviated from the surrogate distribution by more than 2.5 of $z$ score.

**Behavior analysis.** Home cage behavioral phenotypes were classified using published algorithms (Janelia Automatic Animal Behavior Annotator, JAABA)[55] adapted to behavior recognition in side-view videos. We trained behavioral classifiers on a 30-min video recorded on NIRCam-2. To evaluate the classifiers' performance, we selected several random data segments outside of training data, consisted of 50,000 frames in total. Ground-truth labels were provided by investigators independent of this research with three levels of confidences: important, unimportant an unknown. Inter-annotator confusion rates were low for important frames (10.1%), justifying assessment of classifier performance based on important frames (error rates: 7.1%).

To quantitatively measure individual behaviors, *c*ustom graphical user interfaces (GUIs) for individual behaviors were programmed using MATLAB (R2019b, MathWorks). In assessing locomotion, we read in positions of animals on NIRCam-1 ("Track"), and calculated movement time, animal traces, distance traveled & walking speed. In evaluating diversive exploration, we identified in NIRCam-2 videos mouse nose point & tail base ("Rear"), deducted upright truncal extension ($\Delta h$), and estimated rearing onset, offset & duration ($\Delta t$). In determining inspective exploration, we computed mouse anterior–posterior axis in NIRCam-1 videos ("Object"), recognized when the axis were oriented toward & the nose were in the interaction zone (from centroid to 2.5 cm-wide perimeter around an object), and placed a flag on the time of object exploration.

**Immunofluorescence.** We intraperitoneally administered mice with a euthanasia dose of pentobarbital (100 mg/kg body weight). Once in deep anesthesia, animals were perfused with phosphate buffer saline (PBS) followed by cold 4% paraformaldehyde (PFA). Brains were harvested, post-fixed in 4% PFA at 4 °C overnight, cryoprotected in 30% sucrose and sectioned into 50 μm-thick slices on a freezing microtome (Leica SM 2010R). Free-floating sections were washed in PBS, incubated for 1 h in blocking solution (0.3% Triton X-100 & 5% NDS), and kept at 4 °C overnight with primary antibodies (chicken anti-GFP, Aves Labs, 1:4000; rabbit anti-GFAP, Agilent Pathology Solutions, 1:500; mouse anti-S100B, Sigma-Aldrich, 1:400; mouse anti-NeuN, Sigma-Aldrich, 1:400; guinea pig anti-Olig2, Bennett Novitch, 1:20,000) in 0.3% Triton X-100 and 5% NDS. On the day after, sections were washed with PBS, incubated for 2 h at room temperature with secondary antibodies (donkey anti-chicken, Alexa Fluor 488, 1:2000; donkey anti-rabbit, DyLight 650, 1:2000; donkey anti-mouse, Cy™3, 1:2000; donkey anti-guinea pig, Cy™3, 1:2000) in 5% NDS, washed again in PBS, incubated with DAPI (1 μg/ml), mounted on slides and coverslip sealed with mounting medium (Aqua-Poly/Mount, Polysciences #18606-20). Antibody details are shown in Key Resources Table. Epifluorescence Images were taken using Axio Imager.M1 microscope (Carl Zeiss) with a Plan-Apochromat 20x/0.8 M27 objective (420650-9901, Carl Zeiss). Confocal images were acquired using a Zeiss LSM 510 Meta microscope (Carl Zeiss) with an EC Plan-Neofluar 40x/1.30 Oil DIC M27 (420462-9900, Carl Zeiss) objective. We then denoised images using a two-dimensional 3.0 × 3.0 pixels$^2$ Gaussian filter.

**Statistics and reproducibility**
Statistics was performed using R (3.5.1, The R Foundation) and MATLAB (R2019b, MathWorks). We conducted parametric statistics assuming normal distribution and equal variances, except when a significant Shapiro–Wilk test warranted nonparametric statistics. When unequal variances were noted, we accepted results of parametric test, with Brown–Forsythe test reported for the experimental data. In using repeated measures analysis of variance (RM-ANOVA), we applied Greenhouse-Geisser ε correction in the event that Mauchly's tests suggested violation of sphericity. Data were reported as mean ± s.e.m. and 95% confidence interval (CI). $p < 0.05$ was considered significant.

Representative images were selected from multiple replicates. Figure 1d: 9 FOVs from three animals; Fig. 1e: 250 animals; Fig. 5b: 25 animals; Fig. 7b: 12 animals with iAChSnFR $^{Null}$ and 11 with iAChSnFR $^{Active}$.

**Reporting summary**
Further information on research design is available in the Nature Portfolio Reporting Summary linked to this article.

## Data availability
The source data are provided in the Source Data file. Sample images are available in the Figshare repository at https://doi.org/10.6084/m9.figshare.23542803.v1. Larger sets of images are available from the lead contact upon request.

## Code availability
The C# and MATLAB codes used in this paper are available in the GitHub repository at github.com/DEBLab01/NC2023.

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

## Acknowledgements

We thank T. Shelly for machining expertise and members of the Bergles Lab for discussions and comments on the manuscript. These studies were supported by grants from the National Institutes of Health (NS050274; MH100024; and MH083728) and a Discovery Award from The Johns Hopkins University to D.E.B.

## Author contributions

Conceptualization: Y.T.A.G. and D.E.B.; methodology: Y.T.A.G., R.J.C., R.W.P. and J.U.K.; investigation: Y.T.A.G., E.T.H. and R.J.C.; providing reagents: L.L.L. and D.E.B.; formal analysis: Y.T.A.G and E.T.H.; writing—original draft: Y.T.A.G. and D.E.B.; writing—review and editing: Y.T.A.G., E.T.H., R.J.C., R.W.P., L.L.L., J.U.K. and D.E.B.; funding acquisition: J.U.K. and D.E.B.

## Competing interests

The authors declare no competing interests.
