## [Peer Review File · Nature Communications]

Multicore fiber optic imaging reveals that astrocyte calcium activity in the mouse cerebral cortex is modulated by internal motivational stateREVIEWER COMMENTS

Reviewer #1 (Remarks to the Author):

This is the most impressive manuscript I have ever been asked to review. It is a technical and innovative tour-de-force! This team developed a new approach specifically to image astrocyte calcium signals in behaving animals without triggering reactive astrogliosis that would be caused by implanting lenses etc into the brain. Instead they developed a novel strategy to visualize astrocyte calcium signals (and other signals) using their clever fiberscope optical system that they optimized for imaging astrocyte calcium dynamics in vivo.

The perspective they have obtained on astrocyte calcium signals in the awake behaving animal opens broad new areas for astrocyte research.

In particular I point to the section that starts on bottom of page 19:

“We further determined that exposing animals to food, but preventing consumption, initiated responses that were even stronger than if they were allowed to eat; the strongest response was observed in cases where mice were able to smell the resource but could not see or access it. This phenomenon further strengthens the conclusion that cortical astrocyte activity is aligned with contextual salencies, likely reflecting differences in the level of neuromodulatory engagement when goals are predicted but not achieved.”

There are many potential implications in these observations that will transform how we view astrocyte calcium signaling. I have no suggested revisions. This beautiful manuscript is publishable as is and will have a major impact in neuroscience by providing both this original technical approach in addition to the strongest evidence obtained to date linking astrocyte calcium signals to behavioural adaptation.

Reviewer #2 (Remarks to the Author):

This is a very interesting study that provides clear proof of concept data on an improved technique for monitoring astrocyte activity in vivo. The results also confirm previous findings using other techniques and provide novel information about how different behaviors impact astrocyte activity. These sections/comments need attention.

Major comments

1. In various places the authors refer to “circadian” changes in astrocyte activity in their studies. Or refer to their results as measures of “an internal clock”. However, endogenous (circadian) rhythms can only be determined under constant conditions. The longitudinal experiments appear to be performed under a LD cycle, so their results are suggestive of an endogenous rhythm, but not direct evidence of same. The authors should change their wording. “active” and “inactive” periods are terms often used in situations like this.
2. A limitation of the pharmacological experiments is that topical application of various drugs will target all brain cells that bear receptors to the selected pathways (e.g. M1 receptors are also in neurons). Therefore, how can the authors be sure that the changes in astrocytes reflect direct action

on astrocytes, or secondary effects caused by changes in surrounding neurons? I don't see how they can draw strong conclusions without more direct manipulations (e.g. combining topical application with selective manipulation of receptors in astrocytes).

3. Unless this was missed, I don't see much evidence of higher resolution of astrocytic processes (or even subcellular events) in vivo, or how their techniques resolve processes vs. somatic changes in activity in vivo in their various behavioral measures. Most of their in vivo measures are somatic, which is basically what others measured before. They may wish to modify this language if they don't have more data to support it.

Other comments:

1. It would be helpful to define the white dashed lines more clearly in Figure 1E as, I assume, timepoints of forced locomotion.
2. It is unclear to me in Figures 3B and D how the x-axes correspond to lights-on/time-of-day, especially since Mouse A and Mouse B seem to have opposite activity patterns across the 24-h recording period which is concerning.
3. Related to the point above, mice used in behavioral paradigms (e.g., explorations) were tested during the dark phase and exposed to light according to information provided on Page 41 (Quantitative assessment of behaviors). Exposure to light during the dark phase disrupts sleep and circadian rhythms. Were mice used to assess Ca²⁺ across diurnal patterns of light:dark behavior used in the other behavioral paradigms prior to this assessment? If so, that should be stated.
4. For infusion experiments, where was the tip of the cannula placed relative to the cortical surface (i.e., the dorsal-ventral coordinates)?
5. On Page 44, how do the authors define a "full recovery from anesthesia" given that isoflurane affects baseline astroglial calcium signaling.
6. On Page 45, the authors state iChSnFR was injected into GLAST-CreER; Rosa26-*Isl-GCaMP3* mice. Were these mice also treated with tamoxifen? If so, how did the authors differentiate between the GCaMP3 and iChSnFR signals?
7. What time periods were used to calculate F0? For example, in paradigms when there is a pre-and post-manipulation assessment (e.g., novel object), was only the pre-manipulation imaging data used? How was F0 calculated for longitudinal assessments like 24-h diurnal patterns?
8. On Page 17, first paragraph, the authors state citation 29 (Kjaerby et al, 2020, BioRxiv) used miniscopes to image astrocytes, but they used fiber photometry.

Reviewer #3 (Remarks to the Author):

In the present manuscript, Gau et al. present a technique for multi-core high density fiber imaging through a cranial window. This configuration permits exploration of astrocyte dynamics in vivo in freely moving animals. Using this technique combined with behavioural tracking, the authors attempt to reveal how the activity of the astrocytes in V1 of mice is related to different behavioural states of the animal. To this end, they conducted a series of experiments and analyses to associate various behavioural phenomena with neuromodulation and internal states of the animal.

The paper builds on previously published work by the authors, where most of the technology presented here has been shown already. Overall, the technological as well as the neuroscientific advance is rather limited, as laid out below:

Major points:

1) Technology: The technique presented here was already reported by the authors in several publications (some not mentioned in the current manuscript, such as doi:10.1117/12.2038265 from 2014), including: [<https://doi.org/10.1016/j.neuron.2014.04.038>] (cited as a reference [55] in the manuscript). Referring to the 2014 version, the authors state: “Flexible multicore fibers have been previously exploited for photometric analysis [58] and low-resolution, low signal-to-noise ratio detection of fluorescence changes in the brain [55, 59].” However, it is not clear what new conceptual advantages were made in the 2023 version of the system: both 2014 and 2023 versions of the system are equivalent, consisting of the same major optical components. Thus, the system presented here is by far not as novel as the authors make us believe. The only new aspect is an objective mount that enables clamping the miniature objective at the end of the fiber bundle to a chronic window.

Biology: It is not clear why the authors chose the primary visual cortex as the brain area to study how astrocytes exert a “neuromodulatory influence in response to different behaviors and internal states”. What is the role of V1 here?

2) In addition to the point above, some other methods are also outdated and therefore limiting the conclusions that can be drawn from this study. For example, GCaMP3 is far from state-of-the-art and the limitations of its dynamic range limits the sensitivity of the method for detecting more subtle calcium transients in astrocytes. For example, in figure 4 it is not clear if no transients are seen for some of the behavioral sequences because of the low sensitivity of the method. With regard to figure 4C the authors comment: “Note that the most consistent activation occurred at Q-A transitions and during exploration, such as rearing.” In the discussion they write: “Although transient increases in astrocyte calcium levels were occasionally observed during diverse behaviors, Q-A transitions and exploratory behaviors such as rearing were most prominently correlated with enhanced astrocyte activity”. This phrase can be interpreted in a way that all behaviours showed some responses and correlations, but this is not represented in the data. Time-locked responses of astrocytes were observed only for Q-A and exploration behaviours, at least that is what the plot presented shows. Note that “transient increases in astrocyte calcium levels were occasionally observed” would also likely hold true in a time window taken completely randomly without any relation to the behaviour.

This ambiguity could easily be resolved by using state-of-the-art methods, rather than a calcium indicator with limited dynamic range and sensitivity that was released 14 years ago.

Related to these points, does figure 4C show a single exemplary animal or is the presented signal in panel C for a single astrocyte? If it is the case, please provide quantification of all animals/trials/astrocytes. Please, also indicate the standard error of mean on these plots.

3) As was shown in Paukert et al. (2014) and well-reproduced in the present work, astrocyte activation is more related to the transition from Q-A rather than the duration of the active phase. Therefore, it would make sense to plot not only the animal's speed and active time fraction but also to quantify and plot the number of Q-A transitions per hour during circadian variations. Could the difference in astrocyte event probability between active and inactive phases (shown in Figure 3E) be fully or partially explained by the difference in number in Q-A transitions in respective active and inactive phases? The authors should relate Q-A transitions rate to astrocyte event probability to confirm/exclude this possibility.

4) In the abstract, the authors state: “Imaging astrocyte activity during self-initiated behaviors revealed that noradrenergic and cholinergic systems act synergistically to recruit astrocytes during state transitions associated with arousal and attention, which was profoundly modulated by internal state.”. “Synergistically” means that noradrenergic and cholinergic systems act cooperatively, so that there are additive effects. One would expect that the data that supports this conclusion will compare within the same experiment the effect that noradrenergic and cholinergic systems produce acting simultaneously with the effects that are produced when each system acts separately. However, the experiments in Figure 7A only show that local application of antagonists reduces astrocyte responses.

Moreover, evidence for direct action of noradrenaline or acetylcholine on astrocytes is lacking as the locally applied drugs act on all cell types and not specifically on astrocytes only (even if the drug application is restricted to superficial cortical areas). This fact is well-recognized in literature but omitted in the present manuscript. E.g. prazosin will also affect neurons and even blood-vessels and not only astrocytes as α 1-adrenoceptors expressed there too. Thus, the effects shown here can not be directly attributed to astrocytes with the experiments presented by the authors.

5) The authors introduce a procedure to estimate the contribution of the hemodynamics to the GCaMP signal (shown in Figure S1). Using this procedure, they concluded that hemodynamic contribution to the GCaMP signal does exist but its effect on the measurement of calcium transients can be neglected. At the same time, they provide a measurement of the control groups with GFP expressing animals, and conclude that “Q-A transitions did not elicit changes in astrocyte fluorescence”. But the measurement with GFP should also include the effect of hemodynamics, in the same manner as it is present for GCaMP. Please, resolve this contradiction and report the effect of hemodynamic modulation in GFP recordings. A quantification of the hemodynamic response in GFP recordings will also serve as a validation of the procedure that you introduced and validity of the correction to GCaMP signals.

6) The formula in S1 should be properly formatted and the meaning of each term in the formula should be explained to the reader. Is the formula developed by the authors or was it taken from the literature? In the first case please provide a rationale behind this procedure or, alternatively, cite work that describes the procedure and its validity in greater detail.

7) Regarding “Our studies also suggest that α 1 adrenoceptors and group mGluR signalling have contrasting effects on cell synchrony, with the former enhancing astrocyte network synchrony and the latter promoting desynchrony”

Please, provide information on how the SI was calculated. What time window duration was used? What would be the SI matrix for non-classified and randomly sampled matrices? Note that in short (compared to the GCaMP kinetics) time windows low-frequency changes of the calcium levels (global increase or decrease within the time window) will be captured rather than actual synchronisation of the neuronal transients. If it is the case, adjust results and conclusions accordingly.

8) Regarding “Using a new genetically encoded optical sensor for ACh (iAChSnFR) with hundred-millisecond kinetics, we found that this ACh reaches astrocyte membranes when mice explored novel objects” and Figure 7:

The control group with a mutated non-functional iAChSnFR is useful, but the comparison should be also made with respect to the control group with functional iAChSnFR and randomly sampled periods of acquisition. Please also provide the responses of iAChSnFR to Q-A and Rear. Besides, the control for locomotion (arousal state?) of the animal is necessary to show that the effect can not be explained solely by these factors but can robustly be attributed to the particular behaviour.

The example in fig. 7B is not clear. What does the green trace on top represent? When averaging over all events shown in the "Active" panel, there is no consistent change in brightness detectable upon novelty exploration.

9) Figure 8: What is the actual normalisation of the plots? How were the "baselines" for deprived and satiated groups aligned? How was dF/F was quantified in panel C? What is the baseline and how do the authors know when there was minimal activity in a given astrocyte? Please use the same scale for all plots. It seems that there is a consistent response for the Satiated group under all conditions: dF/F does not remain flat, it actually has a negative trend. It is just hidden in the two right panels due to the different scale.

Further, the difference between deprived/satiated states is evident for Chow in panel A, while quantification in panel C shows no significant difference. Is it just a not representative example or does the quantification procedure not grasp this difference?

Minor points:

individual points in Figure 8C are almost invisible for Water and Chow groups.

Figure 7A, what statistical test, $n = ?$. Figure 7 B, what do the purple triangles indicate?

Figure 7A, there is "-1" instead of "1" on the y-scale.

Figure 3B "(red: probability distribution fitted curve)": if some curve is used for fitting, provide a corresponding equation or at least name of the function that was used for the fit.

Reviewer #4 (Remarks to the Author):

Summary:

The authors of the manuscript describe a novel multicore fiber imaging technique to record astrocyte calcium activity in freely moving mice. They then apply this method to map heterogeneous astrocyte calcium activity onto mouse circadian rhythm, spontaneous behavior, novelty exploration, and internal motivational states. Using pharmacological manipulations and fluorescent biosensors, they establish specific contributions of norepinephrine and acetylcholine to the astrocyte calcium signatures. This work makes the important contribution of extending and clarifying the results of previous experiments conducted in situ or in head-fixed in vivo conditions to a freely moving paradigm. Furthermore, their results suggest that known complexity in astrocyte calcium dynamics may be driven by distinct neuromodulators under specific behavioral contexts. Overall, the manuscript is well written, the data is compelling, the analysis is robust, and the conclusions are largely supported by the data.

Critiques:

Though the manuscript contains a substantial amount of data and supporting analysis, it could be improved through consideration of several points relating to additional clarification, data, and/or analysis. The two main components of the paper are the introduction of a new method for astrocyte calcium imaging and subsequent discoveries regarding the nature of astrocyte calcium described using that method. The critiques will be organized as they relate to each component.

Multicore fiber optic imaging

The authors demonstrate a high-density, multi-core fiber coupled to a aspherical lens doublet that provides longitudinal 1-photon imaging of calcium activity in freely moving animals. The most relevant technologies being currently used to study calcium dynamics in freely moving animals are fiber photometry and mini-scope microscopy. The authors argue that their method has improved spatial resolution and decreased inflammatory confounds than these methods. In comparison to fiber photometry, the authors demonstrate that their method allows for spatial resolution of individual astrocytes whereas fiber photometry averages the entire field of view (FOV). However, much heterogeneity of astrocyte calcium events *in vivo* has been documented to occur in the fine processes (i.e., micro-domains) of individual astrocytes. It seems that the multicore fiber optic imaging is able to resolve individual astrocytes but not micro-domains as all the analysis was performed on whole cells. A possible experiment to test this resolution would be to use the IP3R2-knockout mice that have diminished release of intracellular calcium in the somas and primary processes, but retain activity in the micro-domains. If the multi-core fiber optic imaging can resolve the residual activity, then it would be a powerful tool for studying micro-domains *in vivo* as current methods require head-fixation and 2-photon microscopy. At the very least, the authors should discuss the ability or inability of their method to capture the dynamics of micro-domains.

A concern regarding this technique is the problem of photobleaching. Any imaging technique using 1-photon excitation deals with the issue of light scattering and therefore high excitation power is needed to produce enough photons originating from the focal plane to get sufficient signal. The authors indicate that the fibers essentially act as pinholes to restrict noise from excitation outside of the focal plane thereby increasing the spatial resolution but requiring higher power to collect enough signal. This is a principal reason why confocal microscopy is generally ineffective for *in vivo* imaging because bleaching or photodamage occurs when excitation is high enough to get photons from the focal plane through the pinhole. Fiber photometry and mini-scope microscopy mitigate this effect by placing optics or fibers as close to the focal plane as possible, however this method employs a significant working distance (~2.5mm). The authors indicate that photobleaching is a concern as they only image for 1s every 60s for imaging across the circadian cycle. The authors state that they use a 0.75W laser to achieve excitation, but the power at the fiber tip was not described, nor is there any measurement of photobleaching over time. This could be potentially analyzed from the EGFP control sessions in which the change in the static fluorophore emission could be plotted as a function of time of imaging. Regardless, more discussion should be given regarding the issue of photobleaching and/or photodamage induced by 1-photon excitation.

A third question is to what depth in the cortex can the multicore fiber resolve individual astrocytes? Though the 2.5mm working distance allows the lenses to image through the thickness of a cranial window, light scattering in the tissue would be expected to increase significantly. Is this imaging method limited to the dorsal astrocytes of layer 1? Though there may be less injury induced by

imaging through a window rather than an implanted fiber or lens, the relationship between the depth of imaging and the cranial window should be discussed.

One powerful advantage of the method described by the authors is the ability to move the imaging assembly around a cranial window, which could facilitate imaging the same (or different) areas across days. However, the 3D positioning does not seem trivial even with the help of a motorized micro-manipulator. Implanted fibers and mini-scopes have chronic implants that facilitates tracking the same FOV over days, but it seems that the method here requires more precise alignment. It would be useful to see an example of imaging the same FOV across days to demonstrate this longitudinal ability.

Astrocyte calcium dynamics in freely moving mice

The authors demonstrate that astrocyte calcium activity is highest during the active phase of the mouse circadian rhythm (Figure 3), which corroborates previous reports suggesting that astrocyte calcium is coupled to norepinephrine release that is highest during wakefulness. Though the data support this conclusion, I am concerned about the sampling rate of 1s imaging every 60s. It is not clear how astrocyte event frequency compares to the sampling frequency. The higher the astrocyte event frequency, the more likely a lower sampling rate will capture temporal dynamics, which could bias the activity captured during peak activeness. Though I expect the authors will find the same relationship between peak astrocyte calcium event frequency occurring during peak activeness, the degree of difference might be influenced by artifacts in the sampling rate. If the authors collected 1 hour of data in the peak active state and 1 hour in the peak inactive state and compared the astrocyte event frequency between these two extremes with their continuous measurements across the circadian cycle, then they could determine how intermittent sampling compares to continuous sampling and more accurately estimate astrocyte activity across the circadian cycle.

The authors provide compelling evidence demonstrating heterogenous astrocyte activity across different mouse behaviors (Figure 4). Though they demonstrate that head-tethered animals move around an arena with similar speed and distance as when untethered (Figure 2C), they do not provide this comparison for the more sophisticated analyses in Figure 4 where being head-tethered might affect the distribution of more complicated behaviors such as nesting, grooming, or drinking. It would be helpful to compare the distribution of behaviors in untethered animals with tethered animals in this context.

The authors quantify the interesting relationship between mouse rearing and astrocyte calcium activity and conclude that the rise in astrocyte calcium most closely aligns with rearing offset (Figure 5). However, the colored raster in Figure 5G seems to suggest that most astrocyte calcium activity peaks at a set temporal offset (~2.5s) from rearing onset rather than offset. It would be important to find the peak cross-correlation between astrocyte calcium activity and rearing onset and then compare astrocyte correlation to that lag from rearing onset vs. rearing offset. Otherwise, the correlation with rearing offset might be an artifact due to the duration of rearing being 1.5-2.5 seconds long (Figure 5E). This possibility underlies a broader difficulty to mapping the slow calcium dynamics of astrocytes onto behavioral epochs of similar timescales. Despite impressively sophisticated analyses, the resulting claim that astrocyte activation occurs at the end of rearing based on a Gini coefficient difference of 0.24 (onset) and 0.21 (offset) seems to be overly simplistic. Perhaps the authors can robustly estimate the beginning of astrocyte calcium events and compare that timepoint rather than the peaks of the transients, which varies depending on the amplitude

(despite the amplitude not reflecting rearing metrics).

One of the most interesting results in this manuscript is the observation that astrocyte calcium activation in contextual encoding of food is highest when the animal is deprived and can see or smell the food but cannot interact with it, thereby representing an interaction between internal state and the external environment. The authors had previously shown that noradrenergic and cholinergic signaling play distinct roles in driving astrocyte calcium activity in response to different mouse behaviors using targeted pharmacological manipulations. However, they do not extend these manipulations or any other mechanistic manipulation to the fascinating results regarding the food deprivation experiments. It would be very interesting to know how these two neuromodulators contribute (or do not) to the astrocyte calcium events described in Figure 8. Though these results are not necessary to make the manuscript novel and an important contribution to the field, they would strengthen the overall cohesion of the results and discussion.

In summary, I commend the authors for this impressive body of work as it provides both technical and conceptual advancements in astrocyte calcium imaging and represents a significant contribution to the field of astrocyte physiology. The critiques are intended to make this solid manuscript even stronger and more complete.

Response to Reviewers

We thank the reviewers for taking the time to carefully evaluate this manuscript. Your comments and suggestions have improved the study and the clarity of the presentation. We have responded to all questions/concerns below (Reviewer's comments are in bold italics).

Reviewer #1:

This is the most impressive manuscript I have ever been asked to review. It is a technical and innovative tour-de-force! This team developed a new approach specifically to image astrocyte calcium signals in behaving animals without triggering reactive astrogliosis that would be caused by implanting lenses etc into the brain. Instead they developed a novel strategy to visualize astrocyte calcium signals (and other signals) using their clever fiberscope optical system that they optimized for imaging astrocyte calcium dynamics in vivo.

The perspective they have obtained on astrocyte calcium signals in the awake behaving animal opens broad new areas for astrocyte research.

In particular I point to the section that starts on bottom of page 19: "We further determined that exposing animals to food, but preventing consumption, initiated responses that were even stronger than if they were allowed to eat; the strongest response was observed in cases where mice were able to smell the resource but could not see or access it. This phenomenon further strengthens the conclusion that cortical astrocyte activity is aligned with contextual saliencies, likely reflecting differences in the level of neuromodulatory engagement when goals are predicted but not achieved."

There are many potential implications in these observations that will transform how we view astrocyte calcium signaling. I have no suggested revisions. This beautiful manuscript is publishable as is and will have a major impact in neuroscience by providing both this original technical approach in addition to the strongest evidence obtained to date linking astrocyte calcium signals to behavioural adaptation.

Thank you for these kind comments about the experiments and their implications for the field.

Reviewer #2:

This is a very interesting study that provides clear proof of concept data on an improved technique for monitoring astrocyte activity in vivo. The results also confirm previous findings using other techniques and provide novel information about how different behaviors impact astrocyte activity. These sections/comments need attention.

Thank you for commenting about the methodologies and the novelty of the information obtained for the field.

Major comments

1. In various places the authors refer to "circadian" changes in astrocyte activity in their studies. Or refer to their results as measures of "an internal clock". However, endogenous (circadian) rhythms can only be determined under constant conditions. The longitudinal experiments appear to be performed under a LD cycle, so their results are suggestive of an endogenous rhythm, but not direct evidence of same. The authors

should change their wording. “active” and “inactive” periods are terms often used in situations like this.

Thank you for noting this distinction. We have revised the text to refer to active and inactive periods relative to the timing of light exposure (p. 8).

2. A limitation of the pharmacological experiments is that topical application of various drugs will target all brain cells that bear receptors to the selected pathways (e.g. M1 receptors are also in neurons). Therefore, how can the authors be sure that the changes in astrocytes reflect direct action on astrocytes, or secondary effects caused by changes in surrounding neurons? I don’t see how they can draw strong conclusions without more direct manipulations (e.g. combining topical application with selective manipulation of receptors in astrocytes).

We agree with the reviewer that it is not possible to conclude that changes in astrocyte activity in response to local exposure to antagonists arise from selective inhibition of receptors on astrocytes. Our goal with these experiments was to explore whether different neuromodulators contribute to astrocyte activity in different behavioral contexts. Because neurons and astrocytes express many of the same neuromodulatory receptors^{1,2,3}, we have been careful not to overinterpret these findings. We now state this limitation in the **Discussion** section “*Limitations of the study.*” These experiments represent a necessary first step to guide future astrocyte-selective genetic manipulations to probe the involvement of distinct receptor types in these behaviors.

3. Unless this was missed, I don’t see much evidence of higher resolution of astrocytic processes (or even subcellular events) in vivo, or how their techniques resolve processes vs. somatic changes in activity in vivo in their various behavioral measures. Most of their in vivo measures are somatic, which is basically what others measured before. They may wish to modify this language if they don’t have more data to support it.

The benchmarking studies in Figure 1 indicate that under ideal conditions, the multicore fiber imaging platform has the resolution (lateral: 3.5 μm ; radial: 24 μm) to distinguish activity in processes versus the soma of astrocytes, as astrocytes have a diameter of $\sim 85 \mu\text{m}$ in the mouse cortex. However, due to the highly scattering nature of brain tissue and the epifluorescence mode of imaging, the resolution of astrocyte processes is reduced *in vivo*. For this reason, in cases where the activity of individual cells was assessed, fluorescence changes were integrated across the entire cell. We now state this explicitly (p. 22) and have included this aspect of the method in the Discussion section “*Limitations of the study.*”

Other comments:

1. It would be helpful to define the white dashed lines more clearly in Figure 1E as, I assume, timepoints of forced locomotion.

Figure panel 1E has been revised to include this information.

2. It is unclear to me in Figures 3B and D how the x-axes correspond to lights-on/time-of-day, especially since Mouse A and Mouse B seem to have opposite activity patterns across the 24-h recording period which is concerning.

To control for the possible influence of imaging itself on astrocyte activity in these long-term recordings, we began imaging at different times of day. Mouse activity was monitored

simultaneously, allowing us to relate astrocyte calcium activity to the overall activity level of the mice. This aspect of the experimental design has been clarified in the Results (p. 9).

3. Related to the point above, mice used in behavioral paradigms (e.g., explorations) were tested during the dark phase and exposed to light according to information provided on Page 41 (Quantitative assessment of behaviors). Exposure to light during the dark phase disrupts sleep and circadian rhythms. Were mice used to assess Ca²⁺ across diurnal patterns of light:dark behavior used in the other behavioral paradigms prior to this assessment? If so, that should be stated.

As the reviewer notes, behavioral assessments were performed during the dark/active phase of the mice. Mice were not exposed to light during these sequences and stray light emitted through the fiber was minimized with opaque shielding. We did not explicitly test whether astrocyte activity induced during different behavioral paradigms was influenced by the timing of the assessment relative to the diurnal cycle. This is now clarified in the text (p. 47).

4. For infusion experiments, where was the tip of the cannula placed relative to the cortical surface (i.e., the dorsal-ventral coordinates)?

We now note in the Methods that the cannula was placed estimated to be 0.05–0.1 mm above the cortical surface (p. 50).

5. On Page 44, how do the authors define a "full recovery from anesthesia" given that isoflurane affects baseline astroglial calcium signaling?

For these experiments, we reduced the amount of isoflurane to the minimum required to cause light sedation (4% in 1 l/min for induction and 1.5% in 0.5 l/min for maintenance), and waited a minimum of 2 hrs after ceasing exposure. Under these conditions, astrocytes calcium responses to enforced locomotion had the same magnitude as those recorded with longer intervals, suggesting that they had fully recovered.

6. On Page 45, the authors state iAChSnFR was injected into GLAST-CreER; Rosa26-*Isl1*-GCaMP3 mice. Were these mice also treated with tamoxifen? If so, how did the authors differentiate between the GCaMP3 and iAChSnFR signals?

Mice that were injected with AAV.GFAP.AChSnFR were not exposed to tamoxifen to prevent expression of GCaMP3. As the reviewer notes, this would have confounded assessment of iAChSnFR activity due to spectral overlap. This has now been clarified in the Methods (p. 51).

7. What time periods were used to calculate F_0 ? For example, in paradigms when there is a pre-and post-manipulation assessment (e.g., novel object), was only the pre-manipulation imaging data used? How was F_0 calculated for longitudinal assessments like 24-h diurnal patterns?

For behavioral studies, we calculated the baseline F_0 using designated pre- and post-experiment recordings on the same day. These recordings captured the mice at rest, during Q-A transitions, and while roaming, providing a comprehensive view of astrocyte calcium dynamics independent of the behavioral perturbations. Subsequently, as detailed in p. 51, the F_0 was determined based on kernel density estimate.

For longitudinal 24-h, the F_0 was the median fluorescence, as the recording inherently entails diverse calcium changes. This information has now been included on p. 51.

8. On Page 17, first paragraph, the authors state citation 29 (Kjaerby et al, 2020, BioRxiv) used miniscopes to image astrocytes, but they used fiber photometry.

Thank you for catching that incorrect citation. The text has been corrected (p. 18). The citation is also updated to its final published form: *Kjaerby et al, 2022, Nature Neuroscience*.

Reviewer #3:

In the present manuscript, Gau et al. present a technique for multi-core high density fiber imaging through a cranial window. This configuration permits exploration of astrocyte dynamics in vivo in freely moving animals. Using this technique combined with behavioural tracking, the authors attempt to reveal how the activity of the astrocytes in V1 of mice is related to different behavioural states of the animal. To this end, they conducted a series of experiments and analyses to associate various behavioural phenomena with neuromodulation and internal states of the animal. The paper builds on previously published work by the authors, where most of the technology presented here has been shown already. Overall, the technological as well as the neuroscientific advance is rather limited, as laid out below:

Major points:

1) Technology: The technique presented here was already reported by the authors in several publications (some not mentioned in the current manuscript, such as doi:10.1117/12.2038265 from 2014), including: [https://doi.org/10.1016/j.neuron.2014.04.038] (cited as a reference [55] in the manuscript). Referring to the 2014 version, the authors state: “Flexible multicore fibers have been previously exploited for photometric analysis [58] and low-resolution, low signal-to-noise ratio detection of fluorescence changes in the brain [55, 59].” However, it is not clear what new conceptual advantages were made in the 2023 version of the system: both 2014 and 2023 versions of the system are equivalent, consisting of the same major optical components. Thus, the system presented here is by far not as novel as the authors make us believe. The only new aspect is an objective mount that enables clamping the miniature objective at the end of the fiber bundle to a chronic window.

The probes used in the 2014 study were not optimized for imaging and were used exclusively in a non-imaging/photometric configuration. To enable imaging, assembly of the miniature lens objective had to be optimized (Figure 1B). In this study, we rigorously determined the performance of this imaging platform under both ideal and real-world conditions (Figure 1C–E). Assembly of the miniature objective, design of the reproducible head mounting system (Figure 2A, D), assessment of the resolving power of the multicore fiber optic imaging platform, and context-specific demonstration of adequate sensitivity (EGFP vs. GCaMP) are described here for the first time. To clarify, we now include the details of the optimized imaging system that differ from the 2014 photometry configuration on pp. 26 & 60–62.

Biology: It is not clear why the authors chose the primary visual cortex as the brain area to study how astrocytes exert a “neuromodulatory influence in response to different behaviors and internal states”. What is the role of V1 here?

Primary sensory regions are where the contextual information and sensory input integrate, and out of all modalities, the visual system combines ready access and an easy means to manipulate sensory input. Prior studies from our lab and others indicate that astrocytes throughout the cortex are modulated by neurotransmitters^{4,5,6}, with high conservation between cortical regions. As our goal was to focus on neuromodulatory influences and reduce the complexity of direct sensory evoked activity, as would occur in motor or somatosensory areas, we recorded astrocyte activity in the visual system while animals were behaving in controlled lighting. This aspect of the experimental design has been clarified in the Results (p. 6).

2) In addition to the point above, some other methods are also outdated and therefore limiting the conclusions that can be drawn from this study. For example, GCaMP3 is far from state-of-the-art and the limitations of its dynamic range limits the sensitivity of the method for detecting more subtle calcium transients in astrocytes. For example, in figure 4 it is not clear if no transients are seen for some of the behavioral sequences because of the low sensitivity of the method. With regard to figure 4C the authors comment: “Note that the most consistent activation occurred at Q-A transitions and during exploration, such as rearing.” In the discussion they write: “Although transient increases in astrocyte calcium levels were occasionally observed during diverse behaviors, Q-A transitions and exploratory behaviors such as rearing were most prominently correlated with enhanced astrocyte activity”. This phrase can be interpreted in a way that all behaviours showed some responses and correlations, but this is not represented in the data.

Time-locked responses of astrocytes were observed only for Q-A and exploration behaviours, at least that is what the plot presented shows. Note that “transient increases in astrocyte calcium levels were occasionally observed” would also likely hold true in a time window taken completely randomly without any relation to the behaviour.

The data in Figure 4C reveal that transient increases in astrocyte calcium occasionally occurred during all behaviors that were assessed; however, the averages show that consistent changes were only observed during Q-A transitions and during exploration. Under these uncontrolled conditions, with mice freely roaming in the arena, we expect that they would occasionally experience increases in arousal when engaged in a variety of behaviors. Thus, we agree with the reviewer that the biology would suggest that occasional transients would be detected using a randomly assigned time window. We do not feel that these occasional transients are inconsistent with the conclusions made.

This ambiguity could easily be resolved by using state-of-the-art methods, rather than a calcium indicator with limited dynamic range and sensitivity that was released 14 years ago.

We agree, in principle, that our results do not exclude the possibility that there are calcium changes in astrocytes that are not detected using the methods employed in this study. We have been careful not to claim that our results indicate that such potential changes do not exist. More sensitive sensors and imaging methods could always reveal more subtle changes in calcium, changes in calcium in distinct cellular compartments, or fluctuations in other second messengers. We now state this explicitly in the Discussion section “*Limitations of the study.*” We have generated many different mouse lines to enable assessment of astrocyte calcium changes (*R26-IsI-GCaMP3*; *R26-IsI-mGCaMP3*; *R26-IsI-GCaMP6s*; *R26-IsI-mGCaMP6s*)^{6,7,8,9}, and most recently *R26-IsI-GCaMP8s* (unpublished) – our priority has been to generate stable mouse lines rather than use viruses to express these sensors, to reduce inter-animal variability and the

potential for reactive changes. In our experience, there are only marginal differences in the ability to detect astrocyte calcium changes between these lines, with the exception of the membrane targeted variants, which increase the ability to record calcium changes in the fine processes of astrocytes^{7, 10, 11}. GCaMP3 was chosen for these experiments because it has higher baseline fluorescence, which provides advantages for probe alignment and reassessment of astrocyte calcium changes in the same brain regions.

Related to these points, does figure 4C show a single exemplary animal or is the presented signal in panel C for a single astrocyte? If it is the case, please provide quantification of all animals/trials/astrocytes. Please, also indicate the standard error of mean on these plots.

The gray lines shown in Figure 4C represent an average cellular response to individual behavioral events. The red lines represent the average across all 6 animals with standard error of mean, now included as the red shaded region (pp. 33–34).

3) As was shown in Paukert et al. (2014) and well-reproduced in the present work, astrocyte activation is more related to the transition from Q-A rather than the duration of the active phase. Therefore, it would make sense to plot not only the animal's speed and active time fraction but also to quantify and plot the number of Q-A transitions per hour during circadian variations. Could the difference in astrocyte event probability between active and inactive phases (shown in Figure 3E) be fully or partially explained by the difference in number in Q-A transitions in respective active and inactive phases? The authors should relate Q-A transitions rate to astrocyte event probability to confirm/exclude this possibility.

Thank you for this suggestion. We have re-analyzed the data and now include data relating a Q-A transition frequency to astrocyte calcium levels. As shown in the figure below, these events carry a weak, yet statistically significant correlation with astrocyte activity ($r = 0.19$, $\beta = 0.422$, $p < 0.0001$). As the reviewer points out, more frequent sampling would likely enhance this correlation.

4) In the abstract, the authors state: “Imaging astrocyte activity during self-initiated behaviors revealed that noradrenergic and cholinergic systems act synergistically to recruit astrocytes during state transitions associated with arousal and attention, which was profoundly modulated by internal state.”. “Synergistically” means that noradrenergic and cholinergic systems act cooperatively, so that there are additive effects. One would expect that the data that supports this conclusion will compare within the same experiment the effect that noradrenergic and cholinergic systems produce acting simultaneously with the effects that are produced when each system acts separately. However, the experiments in Figure 7A only show that local application of antagonists reduces astrocyte responses.

We used the term synergy to indicate that both neuromodulators were acting at the same time, but agree that this could be misinterpreted to imply cooperativity. We have changed the wording in the text to improve the clarity (p. 2).

Moreover, evidence for direct action of noradrenaline or acetylcholine on astrocytes is lacking as the locally applied drugs act on all cell types and not specifically on astrocytes only (even if the drug application is restricted to superficial cortical areas). This fact is well-recognized in literature but omitted in the present manuscript. E.g. prazosin will also affect neurons and even blood-vessels and not only astrocytes as $\alpha 1$ -adrenoceptors expressed there too. Thus, the effects shown here cannot be directly attributed to astrocytes with the experiments presented by the authors.

As noted above (Reviewer #2, comment #2), we agree that receptors targeted by these antagonists are expressed by multiple cell types. We have been careful to not imply that the effects on astrocyte calcium levels can be ascribed solely to engagement of their receptors. Our goal for these experiments was to explore the involvement of different neuromodulators in the responses of astrocytes to different behaviors. We now discuss this issue in the Discussion section “Limitations of the study”.

5) The authors introduce a procedure to estimate the contribution of the hemodynamics to the GCaMP signal (shown in Figure S1). Using this procedure, they concluded that hemodynamic contribution to the GCaMP signal does exist but its effect on the measurement of calcium transients can be neglected. At the same time, they provide a measurement of the control groups with GFP expressing animals, and conclude that “Q-A transitions did not elicit changes in astrocyte fluorescence”. But the measurement with GFP should also include the effect of hemodynamics, in the same manner as it is present for GCaMP. Please, resolve this contradiction and report the effect of hemodynamic modulation in GFP recordings. A quantification of the hemodynamic response in GFP recordings will also serve as a validation of the procedure that you introduced and validity of the correction to GCaMP signals.

Hemodynamic changes in EGFP fluorescence are not visible in the plots in Figures 1 and 2, because their contribution is negligible at this scale: These changes result in changes of <5% of the amplitude. For reference, an example of original and adjusted plots are shown below.

6) The formula in S1 should be properly formatted and the meaning of each term in the formula should be explained to the reader. Is the formula developed by the authors or was it taken from the literature? In the first case please provide a rationale behind this procedure or, alternatively, cite work that describes the procedure and its validity in greater detail.

We now define all terms in the figure legend. This formula was derived mainly in the following publication: Ma et al. (2016) with various applications^{12, 13, 14, 15, 16}.

7) Regarding “Our studies also suggest that $\alpha 1$ adrenoceptors and group mGluR signaling have contrasting effects on cell synchrony, with the former enhancing astrocyte network synchrony and the latter promoting desynchrony”

Please, provide information on how the SI was calculated. What time window duration was used? What would be the SI matrix for non-classified and randomly sampled matrices? Note that in short (compared to the GCaMP kinetics) time windows low-frequency changes of the calcium levels (global increase or decrease within the time window) will be captured rather than actual synchronization of the neuronal transients. If it is the case, adjust results and conclusions accordingly.

SI was calculated by amplitude-adjusted Fourier transform in conjunction with equal-time correlation analysis, using the 5 s around the behavior of interest as a time interval. We now include the SI for non-classified (0.17 ± 0.01 , 95% CI, [0.15, 0.20], $n = 8$) and random samples (0.20 ± 0.01 , 95% CI, [0.17, 0.22], $n = 8$) for comparison. We did not record neuronal transients in these experiments – all correlation data was computed for astrocytes. These are now included in p. 13.

This amplitude-adjusted Fourier transform plus equal-time correlation analysis is not sensitive to the duration or amplitude of astrocyte events. To ensure the robustness of our synchronization indices, we rigorously tested them against a random time transient series. The process of generating these surrogate data involved using the actual peak transient data to compute time-dependent rates, which were then used to generate surrogate events through an inhomogeneous Poisson process. This procedure was carried out using the amplitude-adjusted Fourier transform from the MATs toolbox, and equal-time correlation analysis was iteratively performed 100 times to create a distribution of surrogate eigenvalues. We considered a result as statistically significant when it deviated by more than 2.5 of z score. We have incorporated the above details into the Methods section (p. 52).

8) Regarding “Using a new genetically encoded optical sensor for ACh (iAChSnFR) with hundred-millisecond kinetics, we found that this ACh reaches astrocyte membranes when mice explored novel objects” and Figure 7:

The control group with a mutated non-functional iAChSnFR is useful, but the comparison should be also made with respect to the control group with functional iAChSnFR and randomly sampled periods of acquisition. Please also provide the responses of iAChSnFR to Q-A and Rear. Besides, the control for locomotion (arousal state?) of the animal is necessary to show that the effect cannot be explained solely by these factors but can robustly be attributed to the particular behaviour.

Thank you for suggesting this additional analysis. The response of iAChSnFR during Q-A, rearing and randomly sampled periods of acquisition is now included (Figure 7C, D, p. 40). No distinct changes are observed in the randomly sampled segments, serving as an internal control. Differential time-varying fluorescence changes are seen around the time of novelty exploration, rearing and Q-A transition. This is consistent with a unique mechanism among behaviors, instead of an umbrella one. These details are now added to Results (p. 15).

The example in fig. 7B is not clear. What does the green trace on top represent? When averaging over all events shown in the “Active” panel, there is no consistent change in brightness detectable upon novelty exploration.

This figure has been revised to include the additional analysis and to improve clarity of the data presentation. The traces above the raster plots in original Figure 7B do not correspond to the raster. These are now displayed separately to avoid confusion (Figure 7B, C).

9) Figure 8: What is the actual normalisation of the plots? How were the “baselines” for deprived and satiated groups aligned? How was dF/F was quantified in panel C? What is the baseline and how do the authors know when there was minimal activity in a given astrocyte? Please use the same scale for all plots. It seems that there is a consistent response for the Satiated group under all conditions: dF/F does not remain flat, it actually has a negative trend. It is just hidden in the two right panels due to the different scale.

We now include additional information for analysis used in this experiment in the Methods: For behavioral studies, we calculated the baseline F_0 using designated pre- and post-experiment recordings on the same day. These recordings captured the mice at rest, during Q-A transitions, and while roaming, providing a comprehensive view of astrocyte calcium dynamics independent of the behavioral perturbations. Subsequently, as detailed in p. 51, the F_0 was determined based on kernel density estimate. The $\Delta F/F$ was then standardized to a time point of 0 within the 5-second window. The data is now presented on a consistent scale.

We agree with the reviewer's observation of the negative trend, which was particularly noticeable when satiated animals encountered the nutritional resource, irrespective of the modality. This was initially observed during spontaneous maintenance behaviors (Figure 4, p. 33) and could indicate a state of contentment. However, in this manuscript, we did not delve into this deeper, as its significance and underlying mechanisms remain uncertain. These intriguing 'cue-inhibited cells' are currently the subject of ongoing investigation. This phenomenon highlights the diverse activities that astrocytes exhibit as they behave in their natural environment, underscoring the strength of the imaging approach.

Further, the difference between deprived/satiated states is evident for Chow in panel A, while quantification in panel C shows no significant difference. Is it just a not representative example or does the quantification procedure not grasp this difference?

A significant difference was not observed when mice were eating the chow. This figure may have been confusing, because of the labeling in upper panel B (labeled “Chow” although this was the “Smell” condition). This figure has been re-labeled for clarity.

Minor points:

individual points in Figure 8C are almost invisible for Water and Chow groups.
These points have been made darker.

Figure 7A, what statistical test, $n = ?$. Figure 7 B, what do the purple triangles indicate?
This information can be found in p. 14 (sample size and detail statistics). Purple triangles represent time when the mice were interacting with novel objects. This has now been defined in the figure legend.

Figure 7A, there is “-1” instead of “1” on the y-scale.
Thank you for catching this error. Corrected.

Figure 3B “(red: probability distribution fitted curve)”: if some curve is used for fitting, provide a corresponding equation or at least name of the function that was used for the fit.

This information has been added to the figure legend.

Reviewer #4:

The authors of the manuscript describe a novel multicore fiber imaging technique to record astrocyte calcium activity in freely moving mice. They then apply this method to map heterogenous astrocyte calcium activity onto mouse circadian rhythm, spontaneous behavior, novelty exploration, and internal motivational states. Using pharmacological manipulations and fluorescent biosensors, they establish specific contributions of norepinephrine and acetylcholine to the astrocyte calcium signatures. This work makes the important contribution of extending and clarifying the results of previous experiments conducted in situ or in head-fixed in vivo conditions to a freely moving paradigm. Furthermore, their results suggest that known complexity in astrocyte calcium dynamics may be driven by distinct neuromodulators under specific behavioral contexts. Overall, the manuscript is well written, the data is compelling, the analysis is robust, and the conclusions are largely supported by the data.

Thank you for noting the robustness of the findings and their importance for the field.

Multicore fiber optic imaging

The authors demonstrate a high-density, multi-core fiber coupled to a aspherical lens doublet that provides longitudinal 1-photon imaging of calcium activity in freely moving animals. The most relevant technologies being currently used to study calcium dynamics in freely moving animals are fiber photometry and mini-scope microscopy. The authors argue that their method has improved spatial resolution and decreased inflammatory confounds than these methods. In comparison to fiber photometry, the authors

demonstrate that their method allows for spatial resolution of individual astrocytes whereas fiber photometry averages the entire field of view (FOV). However, much heterogeneity of astrocyte calcium events in vivo has been documented to occur in the fine processes (i.e., micro-domains) of individual astrocytes. It seems that the multicore fiber optic imaging is able to resolve individual astrocytes but not micro-domains as all the analysis was performed on whole cells. A possible experiment to test this resolution would be to use the IP3R2-knockout mice that have diminished release of intracellular calcium in the somas and primary processes, but retain activity in the micro-domains. If the multi-core fiber optic imaging can resolve the residual activity, then it would be a powerful tool for studying micro-domains in vivo as current methods require head-fixation and 2-photon microscopy. At the very least, the authors should discuss the ability or inability of their method to capture the dynamics of micro-domains.

Thank you for these comments. Although the imaging device could in principle resolve subcellular/microdomain activity, as shown in Figure 1D, the high scattering nature of brain tissue limits unambiguous assessment of fluorescence changes in astrocyte processes. We agree that it would be very exciting to compare microdomain and somatic activity at this broad scale in freely moving animals; however, as we do not maintain the IP3R2 ko mice and detection of activity in these regions would benefit from using membrane anchored rather than cytosolic GCaMP. These experiments would require many rounds of breeding and a delay of more than a year. For this reason, we have revised discussion of this point in the manuscript in the Discussion section “*Limitations of the study.*”

A concern regarding this technique is the problem of photobleaching. Any imaging technique using 1-photon excitation deals with the issue of light scattering and therefore high excitation power is needed to produce enough photons originating from the focal plane to get sufficient signal. The authors indicate that the fibers essentially act as pinholes to restrict noise from excitation outside of the focal plane thereby increasing the spatial resolution but requiring higher power to collect enough signal. This is a principal reason why confocal microscopy is generally ineffective for in vivo imaging because bleaching or photodamage occurs when excitation is high enough to get photons from the focal plane through the pinhole. Fiber photometry and mini-scope microscopy mitigate this effect by placing optics or fibers as close to the focal plane as possible, however this method employs a significant working distance (~2.5mm). The authors indicate that photobleaching is a concern as they only image for 1s every 60s for imaging across the circadian cycle. The authors state that they use a 0.75W laser to achieve excitation, but the power at the fiber tip was not described, nor is there any measurement of photobleaching over time. This could be potentially analyzed from the EGFP control sessions in which the change in the static fluorophore emission could be plotted as a function of time of imaging. Regardless, more discussion should be given regarding the issue of photobleaching and/or photodamage induced by 1-photon excitation.

We agree with the Reviewer about the limitations regarding one photon imaging in brain tissue. We have included additional information about the laser power at the output of the fiber (p. 46) and the rates of photobleaching (0.001 arbitrary unit/min, p. 22). We did not detect evidence of photodamage following illumination, as assessed by an increase in GFAP immunoreactivity and lipofuscin accumulation. As any constant illumination is likely to generate reactive oxygen species and induce damage, we varied the illumination parameters to minimize this complication. We cannot exclude that astrocyte physiology was altered during these studies and

do not claim that other imaging parameters would not induce pathological consequences. These caveats are now addressed in the Discussion section “*Limitations of the study.*”

A third question is to what depth in the cortex can the multicore fiber resolve individual astrocytes? Though the 2.5mm working distance allows the lenses to image through the thickness of a cranial window, light scattering in the tissue would be expected to increase significantly. Is this imaging method limited to the dorsal astrocytes of layer 1? Though there may be less injury induced by imaging through a window rather than an implanted fiber or lens, the relationship between the depth of imaging and the cranial window should be discussed.

The Reviewer is correct that scattering limits the depth that astrocytes can be resolved in the one photon epifluorescence imaging employed here. As a result, all astrocyte activity was recorded from astrocytes ~75 μ m below the cortical surface in Layer I. We now include a discussion about depth imaging in the Results (p. 6) and in the Discussion section “*Limitations of the study.*”

One powerful advantage of the method described by the authors is the ability to move the imaging assembly around a cranial window, which could facilitate imaging the same (or different) areas across days. However, the 3D positioning does not seem trivial even with the help of a motorized micro-manipulator. Implanted fibers and mini-scopes have chronic implants that facilitates tracking the same FOV over days, but it seems that the method here requires more precise alignment. It would be useful to see an example of imaging the same FOV across days to demonstrate this longitudinal ability.

As in other longitudinal imaging studies, we used surface blood vessels as fiducial landmarks to aid in positioning fibers to obtain the same field of view. Examples of repeated imaging of astrocytes are now provided in the Results (Figure S1, p. 63).

Astrocyte calcium dynamics in freely moving mice

The authors demonstrate that astrocyte calcium activity is highest during the active phase of the mouse circadian rhythm (Figure 3), which corroborates previous reports suggesting that astrocyte calcium is coupled to norepinephrine release that is highest during wakefulness. Though the data support this conclusion, I am concerned about the sampling rate of 1s imaging every 60s. It is not clear how astrocyte event frequency compares to the sampling frequency. The higher the astrocyte event frequency, the more likely a lower sampling rate will capture temporal dynamics, which could bias the activity captured during peak activeness. Though I expect the authors will find the same relationship between peak astrocyte calcium event frequency occurring during peak activeness, the degree of difference might be influenced by artifacts in the sampling rate. If the authors collected 1 hour of data in the peak active state and 1 hour in the peak inactive state and compared the astrocyte event frequency between these two extremes with their continuous measurements across the circadian cycle, then they could determine how intermittent sampling compares to continuous sampling and more accurately estimate astrocyte activity across the circadian cycle.

We used a low sampling rate in these experiments to assess the relative activity of cortical astrocytes during active/dark and inactive/light phases. It is possible that the low sampling rate could bias the results, if the duration of each astrocyte event was different between the two

periods. To investigate, we simulated the probability of captured events ($P_{\text{captured event}}$) at different sampling rates (frame per second, fps), using the activeness of the animal, which is also circadian phase-dependent, as a proxy for expected astrocyte event frequency.

Imaging rate (fps)	$P_{\text{captured event}}$ during active phase	$P_{\text{captured event}}$ during inactive phase	Active/inactive ratio
1	4.28%	0.76%	5.63
0.1	4.08%	0.69%	5.91
0.0167	5.14%	0.83%	6.19
0.0083	4.44%	1.11%	4

This analysis demonstrated that, while the undersampling may have caused a slight overestimation of astrocyte activity during the active phase under certain sampling frequencies, it was not necessarily more biased at the lower sampling frequencies and did not alter the fundamental qualitative astrocyte event difference between the two phases.

Overall, this indicates that the low sampling rate should not bias the conclusion that astrocyte activity is greater during the active/dark period.

The authors provide compelling evidence demonstrating heterogenous astrocyte activity across different mouse behaviors (Figure 4). Though they demonstrate that head-tethered animals move around an arena with similar speed and distance as when untethered (Figure 2C), they do not provide this comparison for the more sophisticated analyses in Figure 4 where being head-tethered might affect the distribution of more complicated behaviors such as nesting, grooming, or drinking. It would be helpful to compare the distribution of behaviors in untethered animals with tethered animals in this context.

The result of this analysis is now provided in a new Figure S2 (p. 67) and Results (p. 7). Overall, no distinct differences in distributions were observed.

The authors quantify the interesting relationship between mouse rearing and astrocyte calcium activity and conclude that the rise in astrocyte calcium most closely aligns with rearing offset (Figure 5). However, the colored raster in Figure 5G seems to suggest that most astrocyte calcium activity peaks at a set temporal offset (~2.5s) from rearing onset rather than offset. It would be important to find the peak cross-correlation between astrocyte calcium activity and rearing onset and then compare astrocyte correlation to that lag from rearing onset vs. rearing offset. Otherwise, the correlation with rearing offset might be an artifact due to the duration of rearing being 1.5-2.5 seconds long (Figure 5E). This possibility underlies a broader difficulty to mapping the slow calcium dynamics of astrocytes onto behavioral epochs of similar timescales. Despite impressively sophisticated analyses, the resulting claim that astrocyte activation occurs at the end of rearing based on a Gini coefficient difference of 0.24 (onset) and 0.21 (offset) seems to be overly simplistic. Perhaps the authors can robustly estimate the beginning of astrocyte calcium events and compare that timepoint rather than the peaks of the transients, which varies depending on the amplitude (despite the amplitude not reflecting rearing metrics).

In addition to the statistical analysis performed, which indicate a closer relationship to the offset of rearing, this dependence can be seen in the last series of trials in Figure 5E – when the animals remained longer in the rearing position, the peak of the astrocyte calcium again tracked to the cessation of rearing. If astrocyte calcium levels reach a peak with a fixed temporal offset from the beginning of a rearing event, all peaks should be aligned in this plot, independent of the timing of rearing offset. As suggested by the Reviewer, we repeated the comparison based on onset (10% rise). The result was similar, indicating a closer link to the offset of rearing. This may represent a period of heightened vigilance when animals return their attention to arena and need to make a choice about their next movement. This new analysis (Figure S4, p. 70) and discussion have been added to the manuscript (p. 11).

One of the most interesting results in this manuscript is the observation that astrocyte calcium activation in contextual encoding of food is highest when the animal is deprived and can see or smell the food but cannot interact with it, thereby representing an interaction between internal state and the external environment. The authors had previously shown that noradrenergic and cholinergic signaling play distinct roles in driving astrocyte calcium activity in response to different mouse behaviors using targeted pharmacological manipulations. However, they do not extend these manipulations or any other mechanistic manipulation to the fascinating results regarding the food deprivation experiments. It would be very interesting to know how these two neuromodulators contribute (or do not) to the astrocyte calcium events described in Figure 8. Though these results are not necessary to make the manuscript novel and an important contribution to the field, they would strengthen the overall cohesion of the results and discussion.

We agree that it would be very interesting to explore if this state-dependent astrocyte neuromodulation involves regulation of both norepinephrine and acetylcholine. We hoped to pursue this question, but found that the need for water restriction and invasiveness of the local perfusion method was not compatible with these experiments. After anesthesia to enable local drug application, the mice failed to consistently perform the cue-initiated behaviors. We believe that it will be more rigorous to perform these experiments using astrocyte specific receptor knockout mice. Such studies were planned more than a year ago but have been delayed as a result of pinworm infections in our facility.

In summary, I commend the authors for this impressive body of work as it provides both technical and conceptual advancements in astrocyte calcium imaging and represents a significant contribution to the field of astrocyte physiology. The critiques are intended to make this solid manuscript even stronger and more complete.

Thank you for these insightful comments and critiques.

References cited here:

1. Bayraktar OA, *et al.* Astrocyte layers in the mammalian cerebral cortex revealed by a single-cell in situ transcriptomic map. *Nature neuroscience* **23**, 500-509 (2020).
2. Chai H, *et al.* Neural circuit-specialized astrocytes: transcriptomic, proteomic, morphological, and functional evidence. *Neuron* **95**, 531-549. e539 (2017).

3. Hrvatin S, *et al.* Single-cell analysis of experience-dependent transcriptomic states in the mouse visual cortex. *Nature neuroscience* **21**, 120-129 (2018).
4. Augusto-Oliveira M, *et al.* Astroglia-specific contributions to the regulation of synapses, cognition and behaviour. *Neurosci Biobehav Rev* **118**, 331-357 (2020).
5. Reitman ME, *et al.* Norepinephrine links astrocytic activity to regulation of cortical state. *Nat Neurosci*, (2023).
6. Paukert M, Agarwal A, Cha J, Doze VA, Kang JU, Bergles DE. Norepinephrine controls astroglial responsiveness to local circuit activity. *Neuron* **82**, 1263-1270 (2014).
7. Agarwal A, *et al.* Transient opening of the mitochondrial permeability transition pore induces microdomain calcium transients in astrocyte processes. *Neuron* **93**, 587-605. e587 (2017).
8. Kellner V, Kersbergen CJ, Li S, Babola TA, Saher G, Bergles DE. Dual metabotropic glutamate receptor signaling enables coordination of astrocyte and neuron activity in developing sensory domains. *Neuron* **109**, 2545-2555. e2547 (2021).
9. Lu T-Y, Hanumaihgari P, Hsu ET, Agarwal A, Bergles DE. Norepinephrine enhances oligodendrocyte precursor cell calcium dynamics in the cerebral cortex during arousal. *bioRxiv*, 2022.2008. 2025.505119 (2022).
10. Ye L, Haroon MA, Salinas A, Paukert M. Comparison of GCaMP3 and GCaMP6f for studying astrocyte Ca²⁺ dynamics in the awake mouse brain. *PloS one* **12**, e0181113 (2017).
11. Srinivasan R, *et al.* Ca²⁺ signaling in astrocytes from *Ip3r2*^{-/-} mice in brain slices and during startle responses in vivo. *Nature neuroscience* **18**, 708-717 (2015).
12. Ma Y, *et al.* Resting-state hemodynamics are spatiotemporally coupled to synchronized and symmetric neural activity in excitatory neurons. *Proceedings of the National Academy of Sciences* **113**, E8463-E8471 (2016).
13. Bouchard MB, Chen BR, Burgess SA, Hillman EM. Ultra-fast multispectral optical imaging of cortical oxygenation, blood flow, and intracellular calcium dynamics. *Optics express* **17**, 15670-15678 (2009).
14. Devor A, *et al.* Stimulus-induced changes in blood flow and 2-deoxyglucose uptake dissociate in ipsilateral somatosensory cortex. *Journal of Neuroscience* **28**, 14347-14357 (2008).
15. Hillman EM, *et al.* In vivo optical imaging and dynamic contrast methods for biomedical research. *Philosophical Transactions of the Royal Society A: Mathematical, Physical and Engineering Sciences* **369**, 4620-4643 (2011).
16. Kozberg MG, Ma Y, Shaik MA, Kim SH, Hillman EM. Rapid postnatal expansion of neural networks occurs in an environment of altered neurovascular and neurometabolic coupling. *Journal of Neuroscience* **36**, 6704-6717 (2016).

REVIEWERS' COMMENTS

Reviewer #1 (Remarks to the Author):

The revised manuscript that includes the revisions in response to the reviewers comments is improved from the first initial submission. There are no further points at this time.

Reviewer #2 (Remarks to the Author):

The authors have been very responsive to my concerns. I congratulate them on a very fine study.

Reviewer #3 (Remarks to the Author):

Please see attached file for a direct response to the individual points (red font).

Reviewer #3 (Remarks on code availability):

The link provided here is not working

Reviewer #4 (Remarks to the Author):

The authors have adequately addressed all of my critiques and suggestions.

Reviewer Responses

Reviewer #3:

1) Technology: The technique presented here was already reported by the authors in several publications (some not mentioned in the current manuscript, such as doi:10.1117/12.2038265 from 2014), including: [https://doi.org/10.1016/j.neuron.2014.04.038] (cited as a reference [55] in the manuscript). Referring to the 2014 version, the authors state: “Flexible multicore fibers have been previously exploited for photometric analysis [58] and low-resolution, low signal-to-noise ratio detection of fluorescence changes in the brain [55, 59].” However, it is not clear what new conceptual advantages were made in the 2023 version of the system: both 2014 and 2023 versions of the system are equivalent, consisting of the same major optical components. Thus, the system presented here is by far not as novel as the authors make us believe. The only new aspect is an objective mount that enables clamping the miniature objective at the end of the fiber bundle to a chronic window.

Thank you for raising this issue. The probe/objective assembly used in these previous studies was not optimized for imaging. It was used exclusively in a non-imaging/photometric configuration in the 2014 paper (DOI: 10.1016/j.neuron.2014.04.038) and only photometric traces are shown in this paper. Prior to the current manuscript, it was not clear that this system could be optimized to allow imaging in the freely moving configuration. We did not intend to imply that the design had not been reported previously. Optimizing the objective to allow precise placement of the lenses was not trivial (Figure 1B) and there have been no prior studies assessing the performance of this imaging platform under both ideal and real-world conditions (Figure 1C–E). Assembly of the miniature objective, design of the reproducible head mounting system (Figure 2A, D), assessment of the resolving power of the multicore fiber optic imaging platform, and context-specific demonstration of adequate sensitivity (EGFP vs. GCaMP) are provided here. To clarify this distinction for the reader, we now include the details of the optimized imaging system that differ from the 2014 photometry configuration on pp. 26 & 60–62 and have removed claims about novelty.

The 2014 publication I mentioned above demonstrates imaging through a lens doublet. I do not understand why the authors claim otherwise (probes were used only in photometric configuration). There are several publications by the authors showing imaging (of even astrocytes) in freely moving mice. Already the 2014 paper states in the abstract: “The system uses a flexible endoscopic probe composed of a multi-core coherent fiber-bundle terminated with an approximately 1500-micron working distance objective lens. The fiber-optic neural interface is mounted on a 4-mm² cranial window enabling visualization of glial calcium transients from the same brain region for weeks. We evaluated the system performance through in vivo imaging of GCaMP3 fluorescence in transgenic headrestrained mice during locomotion” In the introduction, the authors then write: “In this work, we present a novel fluorescence fiberoptic microendoscopy approach to detect functional brain activities in a live mice minimally invasively. The system uses a flexible endoscopic probe composed of a 30,000 core, 650-

micron-diameter coherent fiber-bundle terminated with an approximately 1500-micron working distance miniature objective. The fiber-optic neural interface at the distal end of the probe can be mounted to a 4-mm² cranial window without touching brain surface, allowing imaging of locomotion-induced calcium transients in Bergmann glial cells in mice that express the genetically encoded calcium indicator GCaMP3. So far, fiber-optic calcium imaging in awake, moving animals has only been accomplished in neurons with low spatial resolution.^{8,9} To the best of our knowledge, this is the first demonstration of fiber-optic glial calcium imaging in the cerebellum of awake, mobile mice.”

In the discussion, the authors write: “In this work, we have demonstrated a novel fiber-optic microscopy, which allows minimally invasive optical imaging of brain activities evoked by locomotion through in vivo animal studies.” The paper also shows imaging examples. Despite pointing this out in my previous criticism, the authors still ignore this paper in the revised manuscript. Why? As mentioned above, there are additional published studies by the authors that already cover most of the technological development and also use imaging. For example, DOI: 10.3390/app7080858 shows analysis of calcium recordings from astrocytes in freely moving animals. The setup for chronic recordings in freely moving animals has been developed in the following paper: DOI: 10.1117/12.2077034 So, in this manuscript, only marginal updates are made to a system published throughout the last 10 years. Most of the preceding publications of the authors on the iterative development of the technique are not reported here, which I find highly concerning. Once more, by ignoring their previously published body of work, the authors create the false impression in this manuscript that a major technological and conceptual advance underlies their experiments.

Aside from this more general concern, in the revised version it is still not becoming clear how exactly the miniature lens was designed and what the exact improvements of the system were compared to the previously published work. Unfortunately, I could not find any additional information on pp. 26 & 60–62 of the revised manuscript.

Overall, it needs to become clear that the authors built on a previously published method, which they improved for chronic imaging through a chronic window. Claims, such as “To enable visualization of astrocyte calcium changes within the cerebral cortex of freely moving animals, we developed a widefield, long working distance fiber optic imaging system capable of resolving cellular changes in fluorescence emitted by genetically encoded calcium indicators in vivo.” are not justified and exaggerated. They should rather explain more precisely how they improved their widefield, long working distance fiber optic imaging system already reported as early as 2014. Information on the exact design of the lens doublets used for the miniature objective are still not provided (at least, I could not find them).

We have revised the manuscript to note the iterative development of this technology that has made cellular resolution of astrocyte calcium transients during normative behaviors possible.

Regarding our statement “...probes used in the 2014 study were not optimized for imaging and were used exclusively in a non-imaging/photometric configuration...,” we would like to clarify that in the Paukert et al. (2014) paper, we were not able to resolve individual cells during head-fixed imaging; hence, only photometric traces were provided in this study. The previous conference proceedings cited by the reviewer (DOI: 10.3390/app7080858, DOI: 10.1117/12.2077034) provide information for automatically segmenting images collected by the device (images were recorded in head immobilized animals) and details about aligning astrocyte imaging data with

animal movements in an arena, respectively. Note that the lens doublet used here is different from the ball lens used in the 2015 publication (DOI: 10.1117/12.2077034). These publications only provide general information about the experimental configuration and did not rigorously assess performance, as we reported here (Fig. 1 and Supplementary Methods). We agree with the reviewer that this iterative development might not be apparent to the reader as previously written, and have therefore revised the manuscript with specific references incorporated accordingly.

3) As was shown in Paukert et al. (2014) and well-reproduced in the present work, astrocyte activation is more related to the transition from Q-A rather than the duration of the active phase. Therefore, it would make sense to plot not only the animal's speed and active time fraction but also to quantify and plot the number of Q-A transitions per hour during circadian variations. Could the difference in astrocyte event probability between active and inactive phases (shown in Figure 3E) be fully or partially explained by the difference in number in Q-A transitions in respective active and inactive phases? The authors should relate Q-A transitions rate to astrocyte event probability to confirm/exclude this possibility.

Thank you for this suggestion. We have re-analyzed the data and now include data relating a Q-A transition frequency to astrocyte calcium levels. As shown in the figure below, these events carry a weak, yet statistically significant correlation with astrocyte activity ($r = 0.19$, $\beta = 0.422$, $p < 0.0001$). As the reviewer points out, more frequent sampling would likely enhance this correlation.

I cannot find this figure in the manuscript. This is relevant information and should not be omitted from the manuscript.

We again thank the reviewer for the advice and now have the analysis - in a more elaborated form - available with this submission. To account for the relative contribution of sleep-wake cycle and momentary state transitions, we incorporated both terms into the multiple regressions (Supplementary Fig. 5). Consistently, both the extended record of circadian rhythm and episodic Q-A transitions are independently and significantly correlated with astrocyte calcium activity (circadian [0.03, 0.25], Q-A [0.08, 0.47]). Of note, the interaction between Phase*A was not included, due to its statistical insignificance and inferior goodness of fit.

****) Remarks on code availability: The link provided here is not working***

We thank the reviewer for noticing the glitch and have restored the link functionality (github.com/DEBLab01/NC2023).